# Compass-RoPE: Isotropic Rotary Position Embeddings for Vision Transformers

Chengxi Min [1 2]   Wei Wang [1 2]   Yao Zhao [1 2]

## Abstract

Recent works introduce Rotary Position Embeddings (RoPE) into vision transformers (ViTs) to enhance their extrapolation capability, i.e., maintaining performance when inference is conducted on higher resolution images. RoPE encodes positions via rotating phases whose change is controlled by *frequency components*. Strandard 2D RoPE does not generalize well to input resolution changes as it only applies axial frequencies separately along each individual axis. To solve this issue, Mix-RoPE combines xy-axis frequencies, such that it can model position relations in diagonal direction. However, in practice, we observe that the learned 2D frequencies become anisotropic in their direction distributions due to the axial spectral bias in image features, limiting the extrapolation ability of ViTs. Motivated by this observation, we propose Compass-RoPE. We replace the xy cartesian coordinates with a polar parameterization that explicitly decouples frequency scale and angle. By initializing the angle vectors uniformly over $[0, 2\pi)$, it ensures the isotropic direction coverage. Besides, we further introduce discrete Fourier transform (DFT) mixing for the angle vectors, allowing each transformed individual angle vector element to nest multipule angles and thus to enrich angular expressiveness. Extensive experiments on multi-resolution classification and dense prediction tasks show that our Compass-RoPE achieves more stable extrapolation performance under large-scale resolution changes. The code will be available at https://github.com/0930mcx/Compass-RoPE

[1]Institute of Information Science, Beijing Jiaotong University, Beijing 100044, China [2]Visual Intelligence + X International Cooperation Joint Laboratory of the Ministry ofEducation, Beijing 100044, China. Correspondence to: Wei Wang <wei.wang@bjtu.edu.cn>.

*Proceedings of the $43^{rd}$ International Conference on Machine Learning*, Seoul, South Korea. PMLR 306, 2026. Copyright 2026 by the author(s).

## 1. Introduction

Improving extrapolation in vision transformers, i.e., maintaining performance at inference on higher-resolution images than the ones seen during training, has been a long-standing goal. Prior works attribute resolution extrapolation failure to positional modeling, and accordingly they develop more robust positional encodings or attention biases to better accommodate position changes in the token grid (Wu et al., 2021; Chu et al., 2023; Likhomanenko et al., 2021; Press et al., 2022). Representative designs include interpolating absolute 2D embeddings (Dosovitskiy et al., 2021; Hugo et al., 2021), introducing relative position bias (Liu et al., 2021; Wu et al., 2021; Liu et al., 2022), and designing continuous/conditional formulations (Chu et al., 2023; Likhomanenko et al., 2021). However, these methods still become brittle when test token lattice deviates substantially from the training one, leaving *large* resolution extrapolation an open challenge (Ding et al., 2023; Lao et al., 2024).

Rotary Position Embeddings (RoPE) were originally developed for language models and has proven to be effective at generalizing to longer sequences (Su et al., 2024). RoPE employs a phase rotation controlled by a set of *frequency* basis functions to encode positional information. Its variant, 2D RoPE has recently been introduced into vision transformers. Compared to the original absolute position embedding, 2D RoPE shows improved robustness when test image resolution differs from the one during training (Heo et al., 2024). However, 2D RoPE applies axial frequencies independently along the $x$ and $y$ axes, which limits its capability to capture off-axis (e.g., diagonal) relations and leads to degraded performance when resolution shifts. Mix-RoPE, improves upon this by coupling the two axes for richer positional modeling on token and adaptive learning of 2D frequencies (Heo et al., 2024). However, a key degree of freedom remains under-explored: *frequency direction*. A 2D frequency is characterized not only by its scale but also by its orientation, yet how training process affects and whether induces directional bias to these orientations remains unclear.

Fig. 1(a) provides an overview of our observation. By revisiting Mix-RoPE through a frequency–directional lens, we consistently observe that image features in ViT often exhibit an *axial bias* preference in their corresponding spectral energy map. This axial preference co-occur with grid-

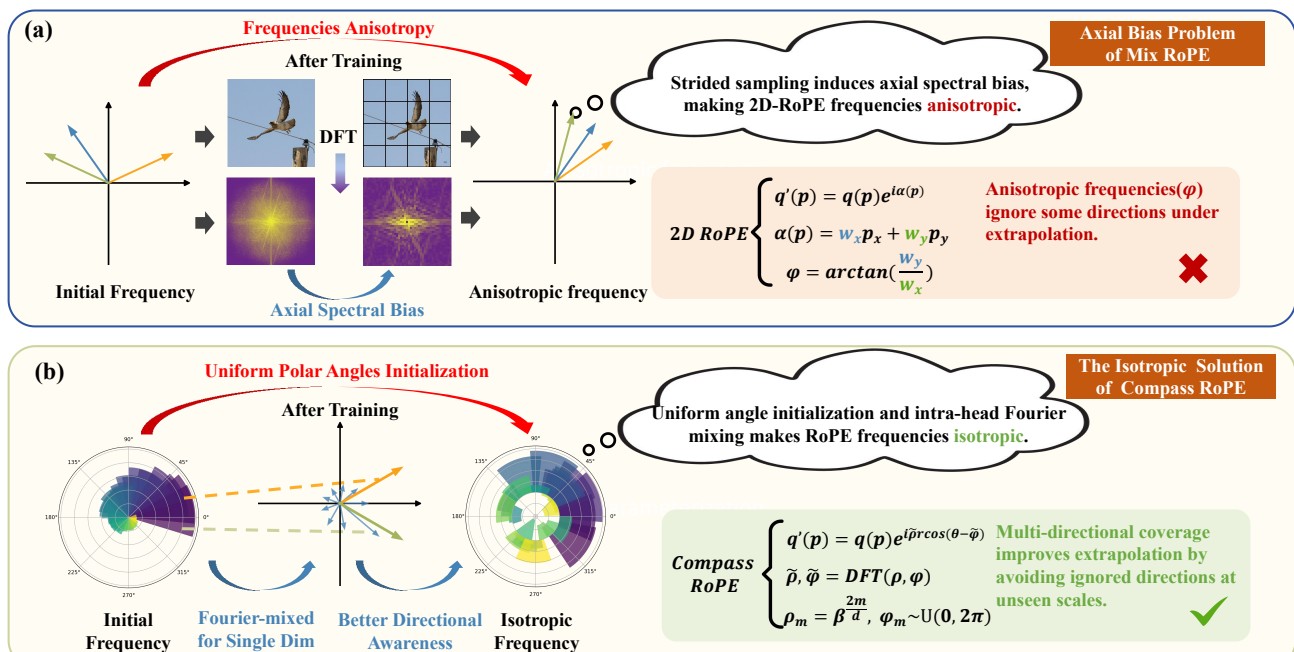

*Figure 1.* **Overview. (a)** Patch tokenization introduces an axial spectral bias in tokenized image patches. This bias skews the optimization of 2D RoPE, encouraging learned frequency orientations to align with the coordinate axes. As a result, the frequency distribution tends to become anisotropic, leading to degraded performance under large resolution shifts. **(b)Compass-RoPE** improves directional coverage by parameterizing each 2D frequency with a scale and an angle, initializing angles uniformly over $[0, 2\pi)$, and applying intra-head DFT mixing so that individual frequency can capture multiple directions.

structured sampling introduced by strided downsampling in patch tokenization process, which converts an image into a sequence of token vectors (Dosovitskiy et al., 2021). From a signal-processing perspective, strided sampling without adequate anti-aliasing operations will distort frequency content (Nyquist, 2002; Shannon, 2006; Zhang, 2019), and recent studies further highlight aliasing issues in transformer-style vision backbones (Michaeli & Soudry, 2025). Such an axial spectrum bias will lead to an anisotropic optimization of RoPE frequency. During the training process, frequency components which aligns with the dominant axes tend to receive stronger and more consistent gradients such that they will also become dominant in the learned frequencies. In contrast, frequency components with oblique orientations which are are driven by weaker feedback are zeroed out (i.e. frequency collapse). This phenomenon is consistent with prior findings that spectral biases can affect model learning during training (Rahaman et al., 2019; Hua et al., 2025). As a consequence, Mix-RoPE frequency will be guided towards *anisotropic directional distribution* (over-allocating capacity to a narrow subset of orientations), which degrades the performance of resolution extrapolation.

Motivated by the above observation, we propose **Compass-RoPE**. Fig. 1(b) summarizes its remedy. We parameterize each 2D frequency in polar form, explicitly decoupling its scale from its orientation. To encourage broad directional coverage from the start of training, we initialize the orien-

tation angles uniformly over $[0, 2\pi)$. To further mitigate directional collapse within each attention head during training, we incorporate intra-head discrete Fourier transform (DFT) mixing (Lee-Thorp et al., 2022) into the RoPE frequency. DFT mixing accommodates individual frequency with multiple directions, enabling it represent multiple orientations—thereby improving directional expressiveness and interactions. Our contributions are three-fold:

- We identify a consistent axial spectral bias in ViT representations and analyze its connection to strided grid-structured sampling (e.g., patch tokenization).
- We propose **Compass-RoPE**, a design that improves directional coverage via polar frequency parameterization with *uniform angle initialization* and intra-head *unitary DFT mixing*.
- We validate Compass-RoPE on ImageNet-1K classification and downstream detection/segmentation tasks, achieving consistent gains in resolution extrapolation. It has superior performance compared with baseline method when training at 224 and evaluating over resolution from the range of 144 to 720.

## 2. Related Work

### 2.1. Positional encodings in vision transformers

ViTs commonly adopt absolute 2D positional embeddings (APE) (Dosovitskiy et al., 2021; Hugo et al., 2021), typi-

cally resized by interpolation under resolution changes. To better capture image locality and improve robustness, many methods have been proposed, including windowed relative position bias (Liu et al., 2021; Wu et al., 2021; Liu et al., 2022), directed-attention designs such as LookHere (Fuller et al., 2024), and conditional/content-aware encodings such as CPE/PEG (Chu et al., 2023; Likhomanenko et al., 2021). Recent work revisits how tokenization/sampling and positional designs affect resolution robustness, e.g., diagnosing shift sensitivity caused by patch embedding and PE (Ding et al., 2023), improving tokenization robustness via sub-token embeddings (Lao et al., 2024), and learning position jointly with content for downstream tasks (Liu et al., 2025).

## 2.2. RoPE and extrapolation-oriented positional modeling in NLP and vision

In NLP, extrapolation-friendly relative-position biases have been extensively studied for length generalization (Shaw et al., 2018; Dai et al., 2019; He et al., 2021; Press et al., 2022). RoPE introduces position-dependent rotations so attention depends on relative offsets via phase differences (Su et al., 2024), and many extensions enable longer-context extrapolation via rescaling or re-parameterization (Sun et al., 2023; Chen et al., 2023; Peng et al., 2024; Ding et al., 2024; Shang et al., 2025). Related frequency-based methods and variants have also been explored, including Fourier Position Embedding (Hua et al., 2025), bilevel positional encoding (He et al., 2024), and higher-dimensional rotary designs (e.g., STRING) (Schenck et al., 2025; Wei et al., 2025).

In vision, learnable 2D RoPE variants aim to improve resolution generalization (Heo et al., 2024; Ostmeier et al., 2024; Yu et al., 2025; van de Geijn et al., 2026), but they largely overlook how tokenization/sampling can bias learned frequency *orientations*. We address this gap by linking strided tokenization to a directional bias in the $q$–$k$ statistics and proposing a polarized RoPE design that improves directional coverage for more stable resolution extrapolation.

## 3. Preliminaries

### 3.1. RoPE for 2D Images

Consider one attention head. Let $\mathbf{p} = (x, y)$ denote an integer token location on the $H_p \times W_p$ patch grid (e.g., $x \in \{0, \dots, W_p{-}1\}, y \in \{0, \dots, H_p{-}1\}$; optionally centered by subtracting the grid center), and let $\mathbf{q}(\mathbf{p}), \mathbf{k}(\mathbf{p}) \in \mathbb{R}^{d_h}$ be the pre-rotation query/key vectors. RoPE groups channels into $D_{\mathrm{pair}} = d_h/2$ two-dimensional (complex) pairs. In 2D RoPE, the first half of the dimensions corresponds to $\omega_x$ and the second half corresponds to $\omega_y$. Specifically, the $m$-th pair is defined as: $\mathbf{q}_m(\mathbf{p}) \in \mathbb{R}^2, \quad \mathbf{k}_m(\mathbf{p}) \in \mathbb{R}^2$. RoPE applies a position-dependent rotation to each pair as

follows:

$$
\begin{aligned}
\mathrm{RoPE}(\mathbf{q}(\mathbf{p}))_m &= \mathbf{R}(\theta_m(\mathbf{p})) \, \mathbf{q}_m(\mathbf{p}), \\
\mathrm{RoPE}(\mathbf{k}(\mathbf{p}))_m &= \mathbf{R}(\theta_m(\mathbf{p})) \, \mathbf{k}_m(\mathbf{p}),
\end{aligned}
\tag{1}
$$

where

$$
\mathbf{R}(\theta) = \begin{bmatrix} \cos\theta & -\sin\theta \\ \sin\theta & \cos\theta \end{bmatrix}.
$$

For 2D images, we compute the phase from a 2D frequency vector by considering the first half of the dimensions as $\omega_x$ and the second half as $\omega_y$, and then concatenate the results:

$$
\theta_m(\mathbf{p}) = [\omega_m^x \cdot x, \omega_m^y \cdot y], \tag{2}
$$

where $\omega_m^x$ corresponds to the frequency in the $x$-direction and $\omega_m^y$ corresponds to the frequency in the $y$-direction.

This formulation gives a relative-position effect: for a displacement $\Delta \in \mathbb{Z}^2$ on the token grid,

$$
\theta_m(\mathbf{p} + \Delta) - \theta_m(\mathbf{p}) = [\omega_m^x \cdot (\Delta_x), \omega_m^y \cdot \Delta_y].
$$

Hence, attention interactions are modulated by frequency-dependent phase differences that depend on the displacement $\Delta$.

### 3.2. Mix-RoPE

Mix-RoPE extends 2D RoPE by coupling the $x$ and $y$ axes for richer positional modeling on token and adaptive learning of 2D frequencies. Concretely, for layer $\ell$ and head $h$, each RoPE pair $m$ is assigned a learnable frequency $\omega_{\ell,h,m} = (\omega_{\ell,h,m}^x, \omega_{\ell,h,m}^y) \in \mathbb{R}^2$, with the corresponding phase

$$
\begin{aligned}
\theta_{\ell,h,m}(\mathbf{p}) &= \omega_{\ell,h,m}^\top \mathbf{p} = \omega_{\ell,h,m}^x \cdot x + \omega_{\ell,h,m}^y \cdot y \\
\theta_{\ell,h,m}(\Delta) &= \omega_{\ell,h,m}^x \cdot \Delta_x + \omega_{\ell,h,m}^y \cdot \Delta_y
\end{aligned}
\tag{3}
$$

By learning both components, Mix-RoPE can capture *off-axis*(e.g. diagonal) relations, and it collapses to an axis-aligned (one-dimensional like 2D RoPE) case when one component is (approximately) zero.

## 4. Axial Spectral Bias Introduces Directional Anisotropy in Mix-RoPE

Token-aligned visual features may exhibit an *axial* bias in the frequency domain. For **learnable** 2D RoPE variants (e.g., Mix-RoPE), this phenomenon has a large impact: since the learnable 2D frequency vectors are updated based on the token features, where axial bias causes gradients to be directionally biased. This results in the learned frequency *orientations* aligning preferentially with few dominate directions, leading to anisotropic distribution and degraded extrapolation performance.

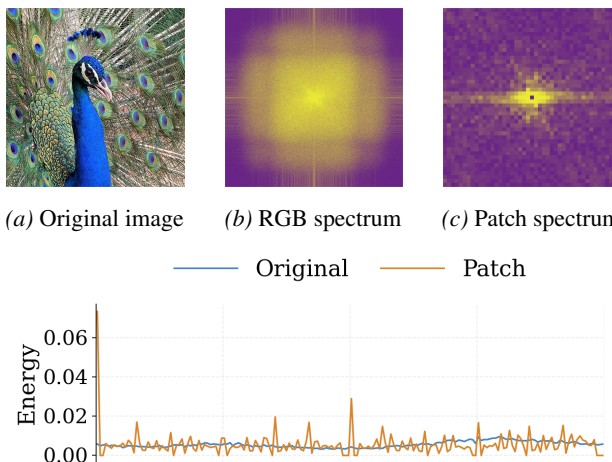

*(a)* Original image     *(b)* RGB spectrum     *(c)* Patch spectrum

*(d)* Angle-energy distribution $E(\theta)$.

*Figure 2.* **Patch tokenization** induces axial structures in frequency. (a) Original RGB image. (b) Power spectrum of the original image. (c) Power spectrum after patch embedding. (d) Angle-energy curves. $E(\theta)$ is computed from Eq. 4.

### 4.1. Axial spectral bias definition

Let $z : \Omega \subset \mathbb{Z}^2 \to \mathbb{R}^C$ be a token-aligned feature map indexed by $p = (x, y)$. We define its channel-averaged power spectrum

$$P_z(\omega) = \frac{1}{C} \sum_{c=1}^{C} \left| \mathcal{F}\{z_c\}(\omega) \right|^2, \qquad \omega = (\omega_x, \omega_y), \quad (4)$$

where $\mathcal{F}$ is the 2D DFT. Using polar coordinates $\omega = (\rho \cos \theta, \rho \sin \theta)$, we say $z$ exhibits *axial spectral bias* if $P_z(\omega)$ distributes near axis-aligned orientations, i.e., $\theta \approx 0$ and/or $\theta \approx \pi/2$.

### 4.2. Patch tokenization is a primary source of axial bias

Patch tokenization in ViT applies a convolution operation to each square region of the input image. The resulting convolutional features are then downsampled onto a regular grid with a spacing of $P$. This process introduces spectral replicas and aliasing along both the horizontal and vertical frequency axes in the spectral domain. These spectral artifacts can then propagate into the token representations, affecting their frequency components of the output features. The deduction process is as follows.

We can view patch tokenization as a process involving convolution followed by down-sampling, represented as:

$$z(p) = \sum_{u \in \mathbb{Z}^2} h(u) \, x(pP - u), \quad (5)$$

where $h$ is the patch kernel (whose size is typically $P \times P$, and it learns a linear projection). In the Fourier domain,

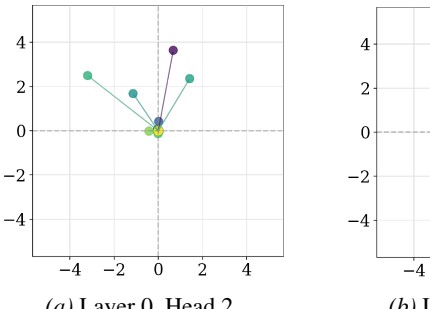

*(a)* Layer 0, Head 2.     *(b)* Layer 8, Head 3.

*Figure 3.* **Orientation anisotropy of learned Mix-RoPE frequencies.** We visualize the learned 2D frequency vectors $\omega$ for a selected layer and head. The distribution of frequency is significantly anisotropic.

strided sampling replicates spectra on a Cartesian grid. According to standard sampling conventions, the resulting spectrum takes the form of

$$Z(\omega) \propto \sum_{k_x, k_y \in \mathbb{Z}} X\left( \frac{\omega + 2\pi(k_x, k_y)}{P} \right) H\left( \frac{\omega + 2\pi(k_x, k_y)}{P} \right), \quad (6)$$

where the Cartesian replication indices $(k_x, k_y)$ correspond to shifts along the $\omega_x$ and $\omega_y$ frequency axes. Because the replication process is separable along the two axes, spectral aliasing tends to preserve—and often enhance—structures aligned with the coordinate axes. In addition, the square spatial structure of $h$ imposes an axis-aligned constraint on its frequency response which can concentrate the spectral energy along the axes, i.e., near orientations $\theta \approx 0$ (horizontal) and $\theta \approx \pi/2$ (vertical).

Empirically, from Figure 2, we can observe that patch-embedded features (Figure 2c) exhibit stronger axis-aligned distribution than original image features (Figure 2b). We further conduct a statistics of energy distribution by an *angle-energy curve* $E(\theta)$, obtained by accumulating spectral energy over a mid-frequency annulus and plotting it with the range of $\theta \in [0, \pi)$.[1] Energy peak near 0 and $\pi/2$ in Figure 2d empirically indicate the axial-aligned distribution in patch-embedd features.

### 4.3. Axial bias induces anisotropic orientations in Mix-RoPE

Mix-RoPE learns 2D frequency vectors $\omega$ (per head) that modulate attention via position-dependent rotations.

Consider one head with pre-rotation $q(p), k(p) \in \mathbb{R}^d$ (with $d$ even), grouped into 2D RoPE pairs $\{q_m(p), k_m(p) \in$

---

[1] Concretely, we sum $P_z(\omega)$ over discrete frequency bins whose radii fall in a mid-band range $\rho \in [\rho_1, \rho_2]$, and then aggregate by the corresponding angle $\theta$ (with angular binning). We restrict to $\theta \in [0, \pi)$ due to the conjugate symmetry of the DFT for real-valued signals.

$\mathbb{R}^2\}_m$. To analyze the gradient dynamics of attention, we introduce an objective function that serves as attention scores during backpropagation. The objective function is defined as:

$$J(\omega) = \sum_\Delta C_{qk}(\Delta)\,\cos(\omega^\top\Delta),$$
$$C_{qk}(\Delta) = \mathbb{E}_p\Big[\sum_m \big\langle q_m(p), k_m(p+\Delta)\big\rangle\Big], \tag{7}$$

where $\Delta$ is a relative displacement on the token grid. Equation (7) isolates the dominant cosine-dependent component from the sine-deppendent component of the RoPE module. The exact RoPE module contains both cosine and sine decomposition with displacement-dependent coefficients; we provide the full form and justify that the sine-related contribution is empirically small in Appendix B. We therefore focus on (7) for the main directional analysis.

Under axial bias, $J(\omega)$ develops anisotropic distribution which aligns along specific orientations in the $\omega$-plane. Importantly, the anisotropic distribution we analyze here does not need remain axis-aligned after the shared $q/k$ projections, as the projection will rotate its orientations. Empirically, we observe a clear directional distribution bias in the head-level $q \cdot k$ spectrum: its energy mostly accumulates in a narrow subset of the angles (see Appendix C), and this is consistent with the analysis above.

To analyze how the anisotropy of $q$–$k$ spatial statistics affects Mix-RoPE update, we analyze how the frequency *orientation* is optimized. Let frequency $\omega = \rho(\cos\phi, \sin\phi)$, we have $\frac{\partial\omega}{\partial\phi} = \rho(-\sin\phi, \cos\phi)$. Let $g = \nabla_\omega J(\omega) = (g_x, g_y)$. According to the chain rule,

$$\frac{\partial J}{\partial\phi} = g^\top\frac{\partial\omega}{\partial\phi} = \rho\,(g_x, g_y)\cdot(-\sin\phi, \cos\phi)$$
$$= \rho\big(-g_x\sin\phi + g_y\cos\phi\big). \tag{8}$$

Equation (8) shows that orientation updates are governed by the *tangential* component of the gradient in the $\omega$-plane.

Let $g = \nabla_\omega J(\omega) = (g_x, g_y)$ and $\psi = \mathrm{atan2}(g_y, g_x)$ denotes the *gradient direction*. Equation (8) then can be reformulated as

$$\frac{\partial J}{\partial\phi} = \rho\|g\|\sin(\psi - \phi), \tag{9}$$

showing that orientation updates are governed by the angular mismatch between $\phi$ and the gradient direction $\psi$. Therefore, if $\psi$ exhibits a persistent directional preference, the frequency orientations $\phi$ are systematically driven toward that direction (up to the $\pi$-periodicity of orientations). Figure 3 visualizes the resulting orientation anisotropy of learned Mix-RoPE frequencies.

Moreover, this orientation anisotropy can directly impair extrapolation performance. When the input resolution changes (and hence the patch grid changes), the distribution of relative displacements $\Delta$ also shifts. If the learned frequencies are directionally anisotropic, Mix-RoPE becomes less responsive to displacements along under-represented directions, weakening attention to direction-specific variations and degrading extrapolation performance.

## 5. Method

We propose **Compass-RoPE** to mitigate anisotropic frequency distributions in Mix RoPE for ViT. Our approach combines polar frequency parameterization with uniform angle initialization and intra-head DFT mixing to enrich angular expressiveness.

### 5.1. Polar Frequency Parameterization with Uniform Initialization

To mitigate axial bias in the learned frequencies, we parameterize each 2D frequency in polar coordinates:

$$\omega_{h,m} = \rho_{h,m}\big(\cos\phi_{h,m},\ \sin\phi_{h,m}\big), \tag{10}$$

where $\rho_{h,m} \in \mathbb{R}$ denotes the radial scale and $\phi_{h,m} \in [0, 2\pi)$ represents the orientation angle. We initialize $\phi_{h,m}$ uniformly over $[0, 2\pi)$ to promote isotropic directional coverage at the beginning of training. The corresponding phase is computed as

$$\theta_{h,m}(p) = \rho_{h,m}\,r(p)\,\cos(\vartheta(p) - \phi_{h,m}), \tag{11}$$

where $(r(p), \vartheta(p))$ are the polar coordinates of the centered token position. For a displacement $\Delta \in \mathbb{Z}^2$ between token $p_1$ and $p_2$ on the token grid, the relative phase shift for the $m$-th frequency component in head $h$ is given by

$$\theta_{h,m}(\Delta) = \rho_{h,m}\cos\phi_{h,m}(r(p_2)\cos\vartheta(p_2) - r(p_1)\cos\vartheta(p_1))$$
$$+ \rho_{h,m}\sin\phi_{h,m}(r(p_2)\sin\vartheta(p_2) - r(p_1)\sin\vartheta(p_1)), \tag{12}$$

which follows the same relative-position encoding formulation as Mix-RoPE in Eq. 3.

### 5.2. Intra-Head DFT Mixing for Enhanced Angular Expressiveness

To further increase directional diversity within each attention head, we apply a unitary DFT mixing across the $D_{\mathrm{pair}}$ frequency pairs. For head $h$, we form complex vectors $z_{h,m} = \rho_{h,m}\exp(i\phi_{h,m})$ and mix them via $\tilde{z}_{h,:} = F z_{h,:}$, where $F$ is the unitary DFT matrix. The mixed parameters $\tilde{\rho}_{h,m} = |\tilde{z}_{h,m}|$ and $\tilde{\phi}_{h,m} = \arg(\tilde{z}_{h,m})$ replace the original ones in Eq. 10. This operation enables each frequency component to nest multiple directional patterns, improving angular expressiveness without additional parameters.

Overall, Compass-RoPE promotes an isotropic distribution of frequencies and enhances their angular expressiveness. The implementation details are provided in Appendix F.

*Table 1.* Multi-resolution evaluation on ImageNet-1K. Models are ViT-Small pre-trained at $224^2$ and evaluated without fine-tuning at different validation resolutions. We report Top-1 accuracy (%). Compass RoPE consistently outperforms all compared methods across all resolutions, with particularly significant gains at higher resolutions.

| Method | 144 | 192 | 224 | 320 | 448 | 512 | 640 | 672 | 720 | AVG. |
|---|---|---|---|---|---|---|---|---|---|---|
| ViT (APE) | 73.6 | 79.1 | 80.4 | 80.6 | 77.6 | 75.4 | 70.3 | 69.1 | 66.8 | 74.8 |
| 2D-RoPE | 73.6 | 79.2 | 80.9 | 81.5 | 78.2 | 76.1 | 67.8 | 65.0 | 59.9 | 73.6 |
| Mix-RoPE | 74.2 | **79.6** | 80.9 | 82.2 | 80.9 | 79.1 | 71.6 | 68.3 | 62.2 | 75.4 |
| Compass-RoPE (Ours) | **74.5** | **79.6** | **81.0** | **82.5** | **81.5** | **80.3** | **75.4** | **73.6** | **70.2** | **77.6** |

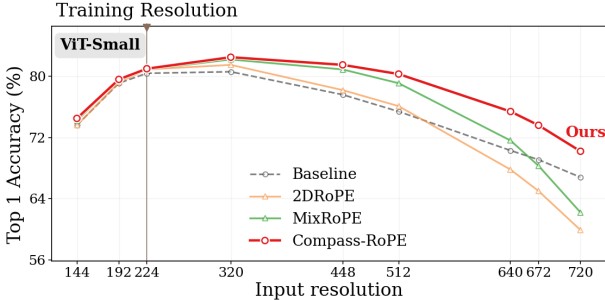

*Figure 4.* Multi-resolution Top-1 accuracy on ImageNet-1K for ViT-S models pre-trained at $224^2$ and evaluated without fine-tuning at varying validation resolutions. The vertical line marks the training resolution. Compass-RoPE improves extrapolation at larger resolutions and remains more stable under resolution shifts.

*Table 2.* Object detection on Pascal VOC with ViT backbones. We report mAP (%) on VOC07 test. Models are trained on VOC07+12 trainval.

| | ViT (APE) | Mix-RoPE | Compass-RoPE (Ours) |
|---|---|---|---|
| mAP (%) | 78.3 | 79.1 | **79.5** |

## 6. Experiments

### 6.1. Experimental setup.

We train our model on ImageNet-1K (Russakovsky et al., 2015) using $8\times$ RTX 4090 GPUs. Unless otherwise specified, we follow the DeiT III training recipe (Touvron et al., 2022) and keep all optimization and data-augmentation settings identical across positional encoding variants for fair comparison. For ViT with absolute positional embeddings (Dosovitskiy et al., 2021), we bicubically interpolate the 2D absolute embeddings to match each evaluation grid, following common practice. Full hyper-parameters and configurations are deferred to the Appendix A.

We additionally evaluate its generalization ability on object detection and semantic segmentation. For detection, we use Faster R-CNN (Ren et al., 2015) on Pascal VOC (Everingham et al., 2010) with VOC07+12 trainval for training and VOC07 test for evaluation. For segmentation, we use UPerNet (Xiao et al., 2018) on ADE20K (Zhou et al., 2017) under the standard protocol. Beyond the main ImageNet-1K ViT-S setting, we further report three supplementary evaluations. These include comparisons with recent baselines(van de Geijn et al., 2026; Ostmeier et al., 2024; Yu et al., 2025; Fuller et al., 2024) and a larger-backbone study on ImageNet-100, as well as evaluation on ImageNet-HR(Fuller et al., 2024). Detailed settings for these supplementary experiments are provided in the corresponding subsections.

### 6.2. Main Experiment: Multi-Resolution Extrapolation for Classification on ImageNet-1K

We evaluate resolution extrapolation on ImageNet-1K (Russakovsky et al., 2015) by comparing ViT with absolute 2D positional embeddings (**ViT (APE)** (Dosovitskiy et al., 2021)), a vanilla 2D rotary baseline from the Mix-RoPE work (**2D-RoPE** (Heo et al., 2024)), **Mix-RoPE** (Heo et al., 2024), and our **Compass-RoPE**. All models are pre-trained at a fixed resolution of $224 \times 224$. During validation, we change only the input resolution without any additional fine-tuning and report Top-1 accuracy at $\{144, 192, 224, 320, 448, 512, 640, 672, 720\}$, along with their average scores across these resolutions (**AVG**). This protocol probes positional encoding behavior under resolution shifts without adaptation, and is aligned with prior resolution-scaling evaluations in ViTs (Liu et al., 2022).

As shown in Table 1 and Fig. 4, Compass-RoPE achieves the best overall accuracy and the most stable extrapolation curve. Its advantage is most pronounced in the high-resolution tail. At the resolution of $640/672/720$, Compass-RoPE reaches $75.4/73.6/70.2$ accuracy, improving over ViT (APE) by $+5.1/+4.5/+3.4$ points. In contrast, Mix-RoPE and 2D-RoPE degrade substantially at these resolutions (Heo et al., 2024), and Mix-RoPE fall below APE at the largest evaluation sizes. Overall, these results support our hypothesis that improving isotropic directional distribution of RoPE frequencies is important for stabilizing RoPE-style positional encoding under large resolution shifts.

### 6.3. Object Detection on Pascal VOC

We also evaluate whether improved extrapolation transfers to the downstream detection task. We use Faster R-CNN (Ren et al., 2015) with ViT backbones and compare three positional encodings: **ViT (APE)**, **Mix-RoPE**, and **Compass-RoPE**. We follow the standard transformer-detector pipelines (Li et al., 2022). For fair comparison, we keep the detector architecture, training schedule, and

*Table 3.* Semantic segmentation on ADE20K with ViT backbones. We report mean IoU (mIoU, %) on the validation set.

|  | ViT (APE) | Mix-RoPE | Compass-RoPE (Ours) |
|---|---|---|---|
| mIoU (%) | 41.1 | 43.2 | **43.4** |

*Table 4.* Ablation on DFT mixing for multi-resolution extrapolation on ImageNet-1K with ViT-Small. We report Top-1 accuracy (%) at selected resolutions. AVG follows Table 1.

| Method | 224 | 448 | 512 | 640 | 720 | AVG |
|---|---|---|---|---|---|---|
| Mix-RoPE | 80.9 | 80.9 | 79.1 | 71.6 | 62.2 | 75.4 |
| Compass w/o DFT | 80.8 | 80.8 | 79.3 | 73.3 | 66.4 | 76.5 |
| Compass-RoPE | **81.0** | **81.5** | **80.3** | **75.4** | **70.2** | **77.6** |

data processing identical and only swap the positional encoding module. Full training details are provided in Appendix A We initialize the ViT backbone from the corresponding ImageNet-1K checkpoint in Table 1. We train on VOC07+12 trainval and evaluate on VOC07 test (Everingham et al., 2010).

Table 2 reports the detection performance on Pascal VOC dataset. We can observe that Compass-RoPE achieves the best mAP among the compared variants. Mix-RoPE improves over the baseline but remains below Compass-RoPE in this setting.

### 6.4. Semantic Segmentation on ADE20K

We evaluate dense prediction on ADE20K (Zhou et al., 2017) using UPerNet (Xiao et al., 2018) with ViT backbones. We compare **ViT (APE)**, **Mix-RoPE**, and **Compass-RoPE**. The segmentation framework, training schedule, and data preprocessing are kept identical across experiments(Appendix A.), with the only variable being the positional encoding design. The ViT backbones are initialized with their corresponding ImageNet-1K pre-trained checkpoints from Table 1.

As shown in Table 3, Compass-RoPE achieves the highest mIoU on ADE20K. While Mix-RoPE improves over the baseline, Compass-RoPE performs best in this comparison.

### 6.5. Ablations on DFT Mixing

We perform an ablation study to assess the impact of the proposed DFT mixing by comparing three variants: **Mix-RoPE**, **Compass-RoPE without DFT**, and **Compass-RoPE**. Compass-RoPE without DFT is identical to the full method except for the removal of DFT mixing. All models are pre-trained at $224^2$ and evaluated at multiple validation resolutions without fine-tuning, following the procedure outlined in Sec. 6.2. For clarity, Table 4 reports results for a subset of the resolutions, and the **AVG** score is computed over all nine evaluated resolutions. The complete table can be found in the Appendix E.

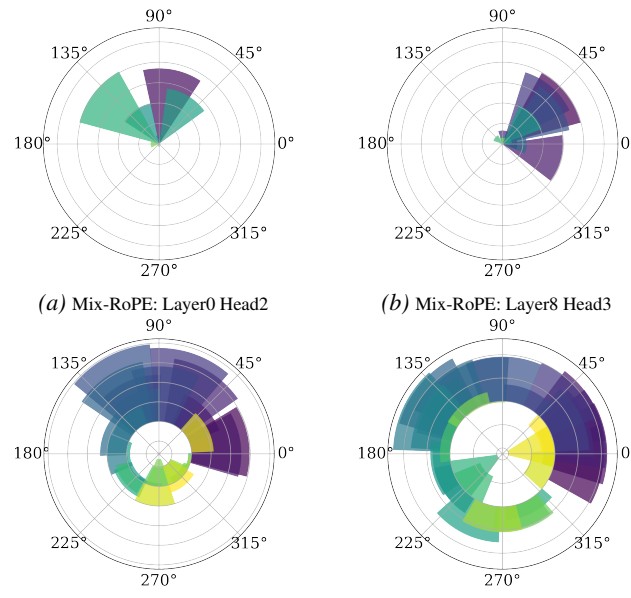

*(a)* Mix-RoPE: Layer0 Head2     *(b)* Mix-RoPE: Layer8 Head3

*(c)* Compass-RoPE: Layer0 Head2     *(d)* Compass-RoPE: Layer8 Head3

*Figure 5.* Qualitative visualization of frequency-direction usage. Each plot shows an energy-weighted angular histogram within one attention head. The angle indicates frequency direction and the radius indicates the accumulated energy in each angular bin. Compared to Mix-RoPE, Compass-RoPE exhibits broader directional spread in both early and deeper layers.

Table 4 shows that DFT mixing further enhances extrapolation performance, with particularly large gains in the high-resolution extremes. Without DFT, Compass-RoPE already outperforms Mix-RoPE at large resolutions ($\geq 512$), but it performs marginally worse near the training resolution ($\leq 448$). We hypothesize that compared to Mix-RoPE's concentration on few dominant directions, the more uniform frequency-direction distribution of Compass-RoPE can increase the initial learning difficulty. However, our complete approach addresses this limitation by incorporating DFT mixing, which enhances the directional expressiveness of individual frequencies. This leads to the best overall AVG score across all evaluated resolutions and the most stable extrapolation curve.

### 6.6. Analysis of Isotropic Frequency

We further validate the superior isotropic frequency distribution of Compass RoPE through three complementary perspectives. First, we quantify its impact on improving the directional coverage of learned 2D frequencies. Second, we investigate whether DFT mixing enhances direction-sensitive correspondence under controlled shifts. Third, we evaluate its robustness under combined rotation and multi-resolution shifts.

**Coverage of learned frequency directions.** We quantify how broadly each method's frequency direction covers. From the checkpoint, we extract and convert the per-head 2D frequency vectors for each RoPE pair into polar form,

*Table 5.* Effective direction count $\exp(H)$ computed from the energy-weighted angular distribution. Compass-RoPE achieves a higher value closer to the upper bound, and thus more uniform directional distribution than Mix-RoPE.

|         | Mix-RoPE | Compass-RoPE (Ours) |
|---------|----------|---------------------|
| Overall | 28.3     | **35.7**            |

*Table 6.* ImagNet-1K validation set's shift alignment accuracy (%, higher is better) under direction-controlled token shifts along 4 directions. The full Compass-RoPE method delivers the most robust performance gains across all directions.

| Method              | 0°       | 45°      | 90°      | 135°     |
|---------------------|----------|----------|----------|----------|
| Mix-RoPE            | 83.9     | 49.1     | 79.8     | 49.9     |
| Compass-RoPE w/o DFT| 85.2     | 57.4     | 82.0     | 60.0     |
| Compass-RoPE (full) | **88.1** | **57.5** | **90.2** | **60.7** |

then build an energy-weighted angular histogram for each head. This is done with 36 bins over $[0, 2\pi)$ by accumulating weights proportional to each vector's squared magnitude. From this distribution, we report the entropy-based effective direction count over all layers and heads, denoted as $\exp(H)$. Detailed definition of $\exp(H)$ can be found in Appendix D.1. Higher values of $\exp(H)$, with a theoretical upper bound of 36, indicate that frequency energy is distributed across a broader range of directions.

Table 5 quantitatively confirms that Compass-RoPE achieves a significantly higher $\exp(H)$ compared to Mix-RoPE, closely approaching the theoretical upper bound. This suggests that its frequency energy is distributed much more uniformly over all directions. The polar plots in Fig. 5 visually validate this enhanced directional coverage.

**Shift alignment accuracy for directional correspondence.** We assess whether DFT mixing improves direction-sensitive correspondence under controlled token-level shifts. Image pairs are constructed by shifting features along four directions $0°, 45°, 90°, 135°$ with known ground-truth displacement. For each direction, we evaluate three shift magnitudes, from 1 to 3 tokens. Given a shifted pair, we perform a local search over candidate displacements and select the best-scoring shift based on an average matching score over valid overlapping token pairs. We report shift alignment accuracy on the validation set of ImageNet-1K as the hit rate, averaged across the three shift magnitudes for each direction, and then averaged over all attention blocks. The metric definition is provided in Appendix D.2.

Table 6 shows that the performance of Mix-RoPE is unstable across directions. Compass-RoPE without DFT partially mitigates the instability in Mix-RoPE, while the full Compass-RoPE method further improves robustness across all directions. Full method achieves the largest gains across all directions, aligning with the goal of DFT mixing, which enhances within-head angular expressiveness by coupling RoPE pairs.

*Table 7.* Rotation robustness on ImageNet-1K under multi-resolution evaluation. Compass-RoPE achieves the most consistent and robust gains across all rotation ranges and resolutions.

| Rotation range | Method | 224 | 512 | 672 | AVG. |
|---|---|---|---|---|---|
| $[-10°, 10°]$ | ViT (APE) | 79.2 | 70.5 | 62.2 | 70.6 |
|  | Mix-RoPE | 79.8 | 75.6 | 60.0 | 71.8 |
|  | Compass-RoPE | **79.9** | **77.4** | **74.6** | **77.3** |
| $[-20°, 20°]$ | ViT (APE) | 78.3 | 66.0 | 55.0 | 66.4 |
|  | Mix-RoPE | 79.1 | 72.6 | 54.6 | 68.8 |
|  | Compass-RoPE | **79.1** | **74.9** | **61.9** | **72.0** |
| $[-30°, 30°]$ | ViT (APE) | 77.1 | 63.9 | 51.4 | 64.1 |
|  | Mix-RoPE | 77.8 | 70.7 | 51.3 | 66.6 |
|  | Compass-RoPE | **78.2** | **73.1** | **58.8** | **70.0** |
| $[-40°, 40°]$ | ViT (APE) | 75.7 | 61.5 | 48.5 | 61.9 |
|  | Mix-RoPE | 76.4 | 69.1 | 49.3 | 64.9 |
|  | Compass-RoPE | **76.5** | **71.4** | **56.2** | **68.0** |

**Rotation robustness under multi-resolution evaluation.** We further evaluate robustness under combined rotation and resolution shifts. The detailed evaluation settings are shown in Appendix D.3. We consider four rotation ranges by sampling the rotation angle $\alpha$ from $[-10°, 10°)$, $[-20°, 20°)$, $[-30°, 30°)$, and $[-40°, 40°)$. For each range, we rotate validation images around the image center using the same interpolation and padding strategy for all methods, and then evaluate Top-1 accuracy at $\{224, 512, 672\}$ without fine-tuning.

Table 7 shows that Compass-RoPE achieves the best accuracy across all rotation ranges and resolutions. It demonstrates a systematic advantage across resolutions, especially at the highest resolution (672), which confirms its superior robustness to combined rotation and scaling. These results confirm that a more isotropic frequency-direction distribution effectively mitigates orientation-related performance degradation.

## 6.7. Additional Results and Discussion

To further strengthen the empirical evidence, we conduct additional experiments involving recent RoPE variants, ImageNet-HR evaluation, and larger ViT backbones.

**Comparisons with Recent Baselines** We additionally compare Compass-RoPE with recent visual RoPE baselines, including **Spherical RoPE** (van de Geijn et al., 2026), **LieRE** (Ostmeier et al., 2024), and **ComRoPE** (Yu et al., 2025), as well as the strong non-RoPE method **LookHere** **(LH)** (Fuller et al., 2024). Following the same protocol, all models are trained on ImageNet-100 at $224^2$ for 200 epochs and evaluated at multiple resolutions.

Table 8 clarifies the relative strengths of these methods. Among recent visual RoPE methods, Compass-RoPE is consistently the strongest across resolutions. LookHere

*Table 8.* Comparison with recent baselines on ImageNet-100. All models are trained at $224^2$ for 200 epochs and evaluated at multiple resolutions without fine-tuning. Compass-RoPE is the strongest among recent visual RoPE baselines. LookHere is stronger at high resolutions, while Compass-RoPE is stronger at low resolutions. Their combination achieves the best overall performance.

| Method | 224 | 448 | 512 | 672 | 720 | AVG. |
|---|---|---|---|---|---|---|
| ViT-Spherical | 73.4 | 74.3 | 73.5 | 68.3 | 66.7 | 71.7 |
| ViT-LieRE | 73.8 | 75.0 | 73.6 | 68.7 | 67.0 | 72.0 |
| ViT-ComRoPE | 74.0 | 75.0 | 73.6 | 69.5 | 67.8 | 72.4 |
| Compass-RoPE | 74.4 | 76.3 | 74.3 | 69.6 | 67.8 | 72.9 |
| ViT-LH | 73.7 | 76.2 | 75.5 | 72.0 | 70.3 | 73.9 |
| Compass-RoPE + LH | **74.6** | **76.9** | **76.0** | **72.4** | **70.4** | **74.3** |

*Table 9.* Evaluation on ImageNet-HR. Compass-RoPE remains superior to APE and Mix-RoPE, showing that its advantage is not an artifact of upsampled validation images.

| Method | 224 | 448 | 512 | 672 | 720 | AVG. |
|---|---|---|---|---|---|---|
| ViT (APE) | 88.4 | 86.2 | 84.5 | 78.8 | 77.2 | 83.3 |
| Mix-RoPE | 88.4 | 88.5 | 86.6 | 77.4 | 72.0 | 83.2 |
| Compass-RoPE | **88.4** | **88.6** | **87.7** | **80.4** | **77.6** | **85.0** |

achieves stronger absolute extrapolation performance in the high-resolution regime, whereas Compass-RoPE is better at lower resolutions. Their combination compensates for the low-resolution weakness of LH while further improving the high-resolution results, leading to the best overall performance. This also shows that Compass-RoPE is not redundant to stronger non-RoPE extrapolation methods, but can serve as an orthogonal improvement.

**Evaluation on ImageNet-HR** To avoid relying on upsampled validation images, we additionally evaluate ImageNet-1K-trained models on **ImageNet-HR** (Fuller et al., 2024), a high-resolution test set.

As shown in Table 9, Compass-RoPE remains the best method on extrapolation resolutions and achieves the highest average performance. This suggests that the gain of Compass-RoPE is robust on a genuine high-resolution dataset, rather than being caused by upsampling protocol.

**Larger-Backbone Validation** We further evaluate a larger backbone by training **ViT-Base** on ImageNet-100 for 200 epochs at $224^2$ and testing at multiple resolutions.

Table 10 shows that the same extrapolation trend persists beyond ViT-Small. Mix-RoPE degrades rapidly at high resolutions and even falls below APE at 720, while Compass-RoPE remains consistently superior. This indicates that the anisotropy/extrapolation issue does not automatically disappear at larger model scale.

**Limitation.** Although Compass-RoPE shows consistent improvements across different backbone scales, our validation on larger backbones is currently conducted on ImageNet-100 rather than the full ImageNet-1K benchmark. Therefore,

*Table 10.* Multi-resolution evaluation on ViT-Base trained on ImageNet-100 for 200 epochs at $224^2$. Compass-RoPE remains consistently stronger than both APE and Mix-RoPE. Notably, Mix-RoPE degrades so severely at 720 that it falls below APE.

| Method | 224 | 512 | 672 | 720 | AVG. |
|---|---|---|---|---|---|
| ViT-Base (APE) | 73.7 | 74.9 | 69.8 | 68.3 | 71.7 |
| ViT-Base (Mix-RoPE) | 76.4 | 77.9 | 70.3 | 67.0 | 72.9 |
| ViT-Base (Compass-RoPE) | **77.7** | **78.2** | **74.7** | **73.7** | **76.1** |

the ViT-Base and ViT-Large results should be interpreted as supplementary evidence for scalability, while full-scale validation on larger backbones remains to be further investigated. This limitation mainly stems from the high computational cost of training large ViT models under the same DeiT-III recipe, especially the requirement of large global batch sizes for stable optimization. Importantly, this limitation reflects the current evaluation scope rather than an inherent restriction of Compass-RoPE itself.

## 7. Conclusion

In this paper, we studied learnable 2D RoPE in Vision Transformers from a frequency-direction perspective and identified an axial spectral bias caused by strided grid-based tokenization, which can lead to anisotropic frequency distributions and unstable resolution extrapolation.

Motivated by this observation, we proposed Compass-RoPE, a polar-coordinate RoPE with uniform angle initialization and intra-head DFT mixing to improve angular expressiveness. Extensive experiments on ImageNet-1K, ImageNet-HR, Pascal VOC, and ADE20K show that Compass-RoPE consistently improves resolution extrapolation and transfers well to downstream tasks. Analyses further verify that these gains are associated with more isotropic directional distributions and stronger robustness under rotation-resolution shifts.

These results highlight directional isotropy as an effective principle for designing vision positional encodings with better extrapolation capability.

## Acknowledgements

This work was supported by the Natural Science Foundation of Beijing, China (Grant No. L252025), and the National Natural Science Foundation of China (No. 62372033, 62120106009).

## Impact Statement

This paper presents work whose goal is to advance the field of machine learning. There are many potential societal consequences of our work, none of which we feel must be specifically highlighted here.

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

# A. Detailed experimental settings

## A.1. ImageNet-1K pre-training.

Unless stated otherwise, we follow the DeiT III training recipe for ViT-S on ImageNet-1K. By default, we train for 400 epochs with global batch size 2048 at resolution $224 \times 224$. We use the LAMB optimizer with cosine learning-rate decay, 5-epoch warmup, weight decay 0.02, and gradient clipping 1.0. The recipe disables RandAugment and uses 3-Augment together with color jitter, horizontal flip, and random resized crop. Mixup and CutMix are enabled with $\alpha = 0.8$ and $\alpha = 1.0$, respectively. At test time, we use crop ratio 1.0. For ViT-S, stochastic depth drop rate is set to 0.0. Full details are summarized in Table 11.

*Table 11.* Training recipe of image classification task on ImageNet-1K.

| Hyperparameter | Value |
|---|---|
| Epochs | 400 |
| Input resolution | $224 \times 224$ |
| Global batch size | 2048 |
| Optimizer | LAMB |
| Base learning rate | $3 \times 10^{-3}$ |
| LR schedule | cosine |
| Warmup epochs | 5 |
| Weight decay | 0.02 |
| Gradient clipping | 1.0 |
| Dropout | 0.0 |
| Stochastic depth (ViT-S) | 0.0 |
| Horizontal flip | enabled |
| Random resized crop | enabled |
| Color jitter | 0.3 |
| RandAugment | disabled |
| 3-Augment | grayscale, gaussian blur, solarization |
| Mixup $\alpha$ | 0.8 |
| CutMix $\alpha$ | 1.0 |
| Test crop ratio | 1.0 |
| Classification loss | BCE |

## A.2. Transfer to detection and segmentation.

For downstream transfer, we keep the training framework fixed and only replace the positional encoding module, while initializing all backbones from the ImageNet-1K pre-trained checkpoints used in the main results. Unless stated otherwise, we use the same random seed across positional encoding variants.

For object detection, we use Faster R-CNN on Pascal VOC, training on VOC07+12 trainval and evaluating on VOC07 test. The detector takes $512 \times 512$ inputs. We train for 24 epochs with AdamW, learning rate $1 \times 10^{-4}$, and a global batch size of 32.

For semantic segmentation, we use UPerNet on ADE20K with standard splits. The input size is $2048 \times 512$ while preserving the original aspect ratio. We train for 12 epochs with AdamW, learning rate $6 \times 10^{-5}$, and a global batch size of 4.

# B. Why the sine term is a small perturbation

We provide the exact RoPE decomposition of the frequency-dependent term and explain why the sine-related contribution (and its induced gradient) is empirically small in our setting.

## B.1. Exact RoPE decomposition: $A(\Delta) \cos(\omega^\top \Delta) + B(\Delta) \sin(\omega^\top \Delta)$

Consider one attention head. Let $q(p), k(p) \in \mathbb{R}^d$ denote the pre-rotation query/key at token position $p$, with $d$ even. RoPE groups channels into 2D pairs, so for the $m$-th pair we write

$$q_m(p) = \begin{bmatrix} q_{2m}(p) \\ q_{2m+1}(p) \end{bmatrix}, \qquad k_m(p) = \begin{bmatrix} k_{2m}(p) \\ k_{2m+1}(p) \end{bmatrix}.$$

Let the rotation angle be $\alpha(p) = \omega^\top p$ for a 2D frequency vector $\omega \in \mathbb{R}^2$. Applying RoPE rotates each pair by $R(\alpha)$, where $R(\alpha)$ is the $2 \times 2$ rotation matrix. The RoPE-modulated dot product for a relative displacement $\Delta$ satisfies

$$\langle R(\alpha(p))\, q_m(p),\ R(\alpha(p+\Delta))\, k_m(p+\Delta)\rangle = q_m(p)^\top R(\alpha(p+\Delta) - \alpha(p))\, k_m(p+\Delta)$$
$$= q_m(p)^\top R(\omega^\top \Delta)\, k_m(p+\Delta). \tag{13}$$

Expanding (13) yields the standard cosine–sine decomposition:

$$q_m(p)^\top R(\omega^\top \Delta)\, k_m(p+\Delta) = \underbrace{\big(q_{2m}(p)k_{2m}(p+\Delta) + q_{2m+1}(p)k_{2m+1}(p+\Delta)\big)}_{\triangleq\, a_m(p,\Delta)} \cos(\omega^\top \Delta)$$
$$+ \underbrace{\big(q_{2m}(p)k_{2m+1}(p+\Delta) - q_{2m+1}(p)k_{2m}(p+\Delta)\big)}_{\triangleq\, b_m(p,\Delta)} \sin(\omega^\top \Delta). \tag{14}$$

Summing over all pairs and averaging over positions defines two displacement statistics:

$$A(\Delta) \;=\; \mathbb{E}_p\Big[\sum_m a_m(p,\Delta)\Big], \qquad B(\Delta) \;=\; \mathbb{E}_p\Big[\sum_m b_m(p,\Delta)\Big]. \tag{15}$$

Therefore, the exact expected frequency-dependent term takes the form

$$J(\omega) \;=\; \sum_\Delta \Big(A(\Delta)\cos(\omega^\top \Delta) \;+\; B(\Delta)\sin(\omega^\top \Delta)\Big). \tag{16}$$

**Interpretation of $A(\Delta)$ and $B(\Delta)$.** $A(\Delta)$ aggregates *aligned* (even channel-even channel and odd channel-odd channel) correlations between $q(p)$ and $k(p+\Delta)$, while $B(\Delta)$ aggregates the *cross* correlations between even and odd channels (even-odd / odd-even) with a signed difference. In complex-number terms, if each pair is viewed as a complex scalar $u_m = q_{2m} + iq_{2m+1}$ and $v_m = k_{2m} + ik_{2m+1}$, then $a_m = \mathrm{Re}(u_m\overline{v_m})$ and $b_m = \mathrm{Im}(u_m\overline{v_m})$. Thus $B(\Delta)$ measures a *quadrature / phase-misalignment* component.

## B.2. Why a small $B(\Delta)$ implies a small sine-gradient contribution

From (16), the gradient w.r.t. $\omega$ decomposes into two parts:

$$\nabla_\omega J(\omega) = \sum_\Delta \Big(- A(\Delta)\sin(\omega^\top \Delta) \;+\; B(\Delta)\cos(\omega^\top \Delta)\Big)\Delta$$
$$\triangleq \nabla_\omega J_A(\omega) \;+\; \nabla_\omega J_B(\omega), \tag{17}$$

where

$$\nabla_\omega J_A(\omega) = \sum_\Delta \big(- A(\Delta)\sin(\omega^\top \Delta)\big)\Delta, \qquad \nabla_\omega J_B(\omega) = \sum_\Delta \big(B(\Delta)\cos(\omega^\top \Delta)\big)\Delta.$$

Since $|\cos(\cdot)| \leq 1$, the $B$-part is directly controlled by the magnitude of $B(\Delta)$:

$$\|\nabla_\omega J_B(\omega)\| \;\leq\; \sum_\Delta |B(\Delta)|\,\|\Delta\|. \tag{18}$$

Likewise, one has $\|\nabla_\omega J_A(\omega)\| \leq \sum_\Delta |A(\Delta)|\,\|\Delta\|$. Hence, when $|B(\Delta)|$ is uniformly small compared to $|A(\Delta)|$ in aggregate, the sine-related gradient $\nabla_\omega J_B$ becomes a *small perturbation* to the dominant cosine-related gradient $\nabla_\omega J_A$. This justifies focusing on the $A(\Delta)\cos(\omega^\top \Delta)$ term for the main directional analysis.

## B.3. Why $B(\Delta)$ is empirically small in ViTs

Equation (15) shows that $B(\Delta)$ is not a generic "odd part" of a scalar correlation; it specifically depends on *cross correlations* between the even and odd channels within each 2D RoPE pair:

$$B(\Delta) = \mathbb{E}_p\Big[\sum_m q_{2m}(p)k_{2m+1}(p+\Delta) \;-\; q_{2m+1}(p)k_{2m}(p+\Delta)\Big].$$

Empirically, we observe that this quadrature component is much smaller than the aligned component

$$A(\Delta) = \mathbb{E}_p\Big[ \sum_m q_{2m}(p)k_{2m}(p+\Delta) + q_{2m+1}(p)k_{2m+1}(p+\Delta) \Big].$$

A plausible explanation is that in standard ViT/DeiT pipelines, queries and keys are produced by linear projections from the same normalized token features (LayerNorm), which tends to yield a covariance structure that is closer to *pair-aligned* (even-even / odd-odd) than *cross-paired* (even-odd / odd-even). Under such near block-diagonal second-order structure within RoPE pairs, the signed cross terms in $B(\Delta)$ are suppressed, while the aligned terms in $A(\Delta)$ remain dominant.

**Empirical validation.** We quantify the relative magnitude of the sine-related term using the *pre-trained* ViT checkpoints and the ImageNet-1K validation set. Averaged over the full validation set, we consistently observe that $B(\Delta)$ is much smaller than $A(\Delta)$, with $\mathbb{E}|B|/\mathbb{E}|A| \approx 0.04$. Under our mid-band evaluation protocol, this translates to a sine-induced orientation-gradient magnitude that is about $20\times$ smaller than its cosine counterpart. These measurements support treating the $B$-weighted sine contribution as a small perturbation in the main derivation.

**Discussion.** While the above provides an operational explanation linking the small sine contribution to the smallness of $B(\Delta)$, a complete characterization of how architectural choices and training dynamics determine $B(\Delta)$ is an interesting direction for future work.

## C. Evidence of anistropic distribution in $q$–$k$ spectra

We provide empirical evidence that the $q$–$k$ correlation spectrum exhibits a stable directional preference. On ImageNet-1K validation images, we extract pre-rotation queries/keys from a given transformer block and head, compute the per-pair cross-spectrum $S_{qk}^{(m)}(\omega) = Q^{(m)}(\omega)\overline{K^{(m)}(\omega)}$, and aggregate its energy over a mid-frequency annulus into an angle-energy curve $E(\theta)$. In a representative setting (block 0, head 0; $14 \times 14$ patch grid; $\rho \in [0.08, 0.30]$ cycles/sample; 36 angle bins), the head-aggregated curve is highly concentrated: the top-1 / top-2 / top-3 / top-5 angle bins account for $43\%$, $62\%$, $79\%$, and $87\%$ of the total energy. These results confirm that $q$–$k$ spatial statistics are directionally anisotropic and dominated by a small number of preferred orientations, which provides a concrete basis for the persistent directional bias assumed in the main analysis.

## D. Additional Analysis Details

This appendix provides the formal definitions and implementation details for the analysis metrics used in Sec. 6.6: (i) the entropy-based effective direction count $\exp(H)$ for frequency-direction coverage, and (ii) the shift alignment accuracy for direction-sensitive correspondence.

### D.1. Effective Direction Count for Frequency-Direction Coverage

**Frequency vectors.** For a given layer $\ell$ and attention head $h$, RoPE-style methods maintain a set of 2D frequency vectors $\mathbf{f}_{\ell,h,m} \in \mathbb{R}^2$ indexed by the RoPE pair $m$:

$$\mathbf{f}_{\ell,h,m} = \big(f_{\ell,h,m}^x,\ f_{\ell,h,m}^y\big). \tag{19}$$

We extract these vectors directly from the model checkpoint. (For methods that share frequencies across layers/heads, the same definition applies with the corresponding indexing.)

**Polar conversion and energy weights.** Each frequency vector is converted into polar coordinates, with direction angle and magnitude defined as

$$\theta_{\ell,h,m} = \text{atan2}\Big(f_{\ell,h,m}^y,\ f_{\ell,h,m}^x\Big) \in (-\pi, \pi], \qquad r_{\ell,h,m} = \|\mathbf{f}_{\ell,h,m}\|_2. \tag{20}$$

We use an energy weight $w_{\ell,h,m} = r_{\ell,h,m}^2$ to emphasize frequency components with larger magnitude.

**Angular histogram with $B = 36$ bins.** We map angles to $[0, 2\pi)$ by $\tilde{\theta} = (\theta + 2\pi) \bmod 2\pi$ and discretize directions into $B{=}36$ uniform bins over $[0, 2\pi)$. Let $\text{bin}(\tilde{\theta}) \in \{1, \ldots, B\}$ denote the bin index. For each head, we accumulate weighted

mass into bins:

$$u_{\ell,h,b} = \sum_{m:\,\text{bin}(\tilde{\theta}_{\ell,h,m})=b} w_{\ell,h,m}, \tag{21}$$

and normalize to obtain a discrete distribution $p_{\ell,h}$:

$$p_{\ell,h,b} = \frac{u_{\ell,h,b}}{\sum_{b'=1}^{B} u_{\ell,h,b'}}, \qquad \sum_{b=1}^{B} p_{\ell,h,b} = 1. \tag{22}$$

**Entropy and effective direction count.** We compute the entropy of the direction distribution for each head,

$$H_{\ell,h} = -\sum_{b=1}^{B} p_{\ell,h,b} \log p_{\ell,h,b}, \tag{23}$$

and report the entropy-based effective direction count

$$N_{\text{eff}}(\ell,h) = \exp(H_{\ell,h}). \tag{24}$$

$N_{\text{eff}}$ increases when energy is spread across more directions and decreases when it concentrates on a few angles. In the main paper (Table 5), we report the overall average across all layers and heads:

$$\exp(H) = \frac{1}{|\mathcal{L}||\mathcal{H}|} \sum_{\ell \in \mathcal{L}} \sum_{h \in \mathcal{H}} N_{\text{eff}}(\ell,h). \tag{25}$$

### D.2. Shift Alignment Accuracy for Direction-Sensitive Correspondence

**Shift directions and magnitudes.** We evaluate token-level shifts along four canonical directions $\{0°, 45°, 90°, 135°\}$. For each direction, we test three magnitudes in token units: $\{1, 2, 3\}$. A direction corresponds to a unit displacement vector on the token grid (e.g., right, down-right, down, down-left), and the ground-truth shift is the direction vector multiplied by the chosen magnitude.

**Feature extraction.** For a chosen attention block (layer), we extract the post-encoding query and key features from the attention module. We discard the class token and reshape the remaining spatial tokens into a 2D grid. Unless otherwise stated, we average scores across heads; this yields a single query/key feature map per block.

**Local search and matching score.** Given an original image and its shifted counterpart, we perform a local search over candidate displacements $\delta$ in the same range as the evaluated magnitudes. For each candidate $\delta$, we align the query map $\mathbf{q}$ and the shifted key map $\mathbf{k}$ by comparing over valid overlapping token positions only. We compute the matching score by averaging cosine similarity over the overlap region:

$$S(\delta) = \frac{1}{|\Omega_\delta|} \sum_{i \in \Omega_\delta} \frac{\langle \mathbf{q}(i), \mathbf{k}(i+\delta) \rangle}{\|\mathbf{q}(i)\|_2 \, \|\mathbf{k}(i+\delta)\|_2}, \tag{26}$$

where $\Omega_\delta$ indexes spatial tokens whose shifted locations remain within bounds. The predicted displacement is the maximizer

$$\hat{\Delta} = \arg\max_\delta S(\delta). \tag{27}$$

**Accuracy aggregation.** For each trial with ground-truth shift $\Delta$, we mark a hit if $\hat{\Delta} = \Delta$. For each direction, we average the hit rate over magnitudes $\{1, 2, 3\}$. In the main paper (Table 6), we report per-direction accuracy and average the accuracy over all attention blocks (all layers).

**Image sampling and repeats.** We compute the reported accuracy on the ImageNet-1K validation set. To reduce sampling variance, each setting is repeated three times with independently sampled images, and the final numbers are averaged across repeats. All methods use the same sampling strategy, shift magnitudes, and search range for fair comparison.

*Table 12.* Ablation on DFT mixing for multi-resolution extrapolation on ImageNet-1K with ViT-Small. We report Top-1 accuracy (%) at 9 resolutions.

| Method | 144 | 192 | 224 | 448 | 512 | 640 | 672 | 720 | AVG |
|---|---|---|---|---|---|---|---|---|---|
| Mix-RoPE | 74.2 | **79.6** | 80.9 | 80.9 | 79.1 | 71.6 | 68.3 | 62.2 | 75.4 |
| Compass w/o DFT | 74.1 | 79.3 | 80.8 | 80.8 | 79.3 | 73.3 | 70.8 | 66.4 | 76.3 |
| Compass-RoPE | **74.5** | **79.6** | **81.0** | **81.5** | **80.3** | **75.4** | **70.2** | **70.2** | **77.6** |

### D.3. Rotation robustness evaluation details

For the rotation robustness test, we apply image rotations using `torchvision.transforms.RandomRotation`. We fix the random seed to `0` to ensure reproducibility, and evaluate with a batch size of `4`. We use the default `torchvision` settings for interpolation and fill values.

# E. Complete Table for Ablation Study

The complete table of ablation study on 9 resolution is shown in Table 12.

# F. Implementation Details

**Parameterization and tensor shapes.** We store Compass-RoPE frequencies as two real tensors per attention block: a radial scale parameter $\rho$ and an orientation angle $\phi$. In code, we pack them into `freqs` of shape $[2, \text{depth}, D]$, where $D = H \cdot D_{\text{pair}}$, $H$ is the number of heads, and $D_{\text{pair}} = d_h/2$ is the number of RoPE pairs per head. We reshape `freqs` into $[2, \text{depth}, H, D_{\text{pair}}]$ when computing phases.

**No positivity constraint on $\rho$.** We do not enforce $\rho \geq 0$ in either the parameterization or the mixing stage. This is because the polar frequency vector in Eq. (10) is invariant to the reparameterization $(\rho, \phi) \equiv (-\rho, \phi + \pi)$, i.e., the sign of $\rho$ can always be absorbed into a $\pi$ shift of $\phi$. Allowing $\rho$ to take signed values therefore does not reduce expressiveness, and simplifies optimization.

**Numerical details for $(r, \vartheta)$ and centering.** To avoid the undefined case $\text{atan2}(0, 0)$, we choose the grid center $(c_x, c_y)$ to be a half-integer offset from integer token indices. Concretely, for a patch grid of width $W_p$ and height $H_p$ with integer indices $x \in \{0, \ldots, W_p - 1\}$ and $y \in \{0, \ldots, H_p - 1\}$, we set

$$c_x = \frac{W_p}{2} - \frac{1}{2} \cdot \mathbb{1}[W_p \text{ is even}], \qquad c_y = \frac{H_p}{2} - \frac{1}{2} \cdot \mathbb{1}[H_p \text{ is even}].$$

For example, for a $14 \times 14$ grid we use $(c_x, c_y) = (6.5, 6.5)$, and for a $15 \times 15$ grid we use $(7.5, 7.5)$. Since $(x, y)$ are integers, $(x - c_x, y - c_y)$ can never be exactly $(0, 0)$ under this choice, and thus $\vartheta(p)$ is well-defined for all tokens. We additionally add a small $\epsilon$ to $r(p)$ for numerical stability.

**Polar positional coordinates and phase construction.** Given token coordinates $(x, y)$ on the patch grid, we first center them by $(c_x, c_y)$ and convert to polar coordinates

$$r(p) = \sqrt{(x - c_x)^2 + (y - c_y)^2} + \epsilon, \qquad \vartheta(p) = \text{atan2}(y - c_y, \, x - c_x).$$

For layer $\ell$, head $h$, and pair index $m$, the Compass-RoPE phase is computed as

$$\theta_{\ell, h, m}(p) = r(p) \, \rho_{\ell, h, m} \, \cos\big(\vartheta(p) - \phi_{\ell, h, m}\big).$$

We then form the complex rotation factors $\text{cis}_{\ell, h, m}(p) = \exp\big(i \, \theta_{\ell, h, m}(p)\big)$, which are used to rotate query and key pairs in the RoPE operator.

**Intra-head unitary DFT mixing.** To prevent within-head directional collapse, we optionally mix the per-head RoPE pairs along the pair dimension. For each layer $\ell$ and head $h$, we first form complex phasors

$$z_{\ell, h, m} = \rho_{\ell, h, m} \exp(i \phi_{\ell, h, m}), \qquad m = 1, \ldots, D_{\text{pair}}.$$

---

**Algorithm 1** Compass-RoPE: polar phase and optional DFT mixing

---

**Input:** `freqs` with $\rho, \phi$ in shape $[2, L, D]$, token coords $(x_n, y_n)_{n=1}^N$, center $(c_x, c_y)$, head count $H$, flag `use_dft`
**Output:** `freqs_cis` in shape $[L, H, N, D_{\text{pair}}]$

1. Reshape `freqs` to $\rho, \phi$ in shape $[L, H, D_{\text{pair}}]$ where $D_{\text{pair}} = D/H$

2. If `use_dft` is true, for each layer $\ell$ and head $h$

   (a) Form $z_{\ell,h,m} = \rho_{\ell,h,m} \exp(i\phi_{\ell,h,m})$ for $m = 1, \ldots, D_{\text{pair}}$
   (b) Compute $\tilde{z}_{\ell,h,:} = \text{DFT}(z_{\ell,h,:})$ using a unitary DFT
   (c) Set $\rho_{\ell,h,m} \leftarrow |\tilde{z}_{\ell,h,m}|$, $\phi_{\ell,h,m} \leftarrow \arg(\tilde{z}_{\ell,h,m})$

3. Center positions: $\bar{x}_n = x_n - c_x$, $\bar{y}_n = y_n - c_y$

4. Convert to polar: $r_n = \sqrt{\bar{x}_n^2 + \bar{y}_n^2} + \epsilon$, $\vartheta_n = \text{atan2}(\bar{y}_n, \bar{x}_n)$

5. For each layer $\ell$, head $h$, pair $m$, token $n$, compute phase

$$\theta_{\ell,h,n,m} = r_n\, \rho_{\ell,h,m}\, \cos(\vartheta_n - \phi_{\ell,h,m})$$

6. Output $\texttt{freqs\_cis}_{\ell,h,n,m} = \exp(i\, \theta_{\ell,h,n,m})$

---

*Table 13.* Multi-resolution evaluation on Swin-Tiny. Compass-RoPE remains stronger than both the original RPB and Mix-RoPE at high resolutions.

| Method | 224 | 512 | 672 | 720 | AVG. |
|---|---|---|---|---|---|
| Swin-Tiny (RPB) | 81.2 | 77.3 | 75.1 | 69.3 | 75.7 |
| Swin-Tiny (Mix-RoPE) | **81.4** | 77.4 | 74.9 | 65.2 | 74.8 |
| Swin-Tiny (Compass-RoPE) | **81.4** | **78.3** | **76.3** | **70.3** | **76.6** |

We apply the unitary DFT along $m$,

$$\tilde{z}_{\ell,h,:} = \mathbf{F}\, z_{\ell,h,:}, \qquad \mathbf{F}_{u,v} = \frac{1}{\sqrt{D_{\text{pair}}}} \exp\left(-2\pi i\, \frac{uv}{D_{\text{pair}}}\right),$$

and convert back to polar form $\tilde{\rho}_{\ell,h,m} = |\tilde{z}_{\ell,h,m}|$ and $\tilde{\phi}_{\ell,h,m} = \arg(\tilde{z}_{\ell,h,m})$. In implementation, this is realized by `torch.fft.fft` with `norm=ortho`, which is equivalent to the above unitary DFT matrix multiplication. The mixed parameters $\tilde{\rho}, \tilde{\phi}$ are then used in the phase formula.

**Initialization.** We initialize the radial scale $\rho$ using a RoPE-style geometric sequence over pair index $m$, and perturb it slightly by a small uniform noise. We initialize $\phi$ to cover the full angular range $[0, 2\pi)$. Concretely, we partition $[0, 2\pi)$ into $D_{\text{pair}}$ equal-width bins and sample one angle uniformly from each bin, independently per head. This yields near-uniform directional coverage at the start of training.

**Precision note.** We compute phases and complex exponentials in full precision by disabling mixed-precision autocast in this module, as the phase values can be sensitive to reduced precision.

**Pseudo-code** The pseudo-code of the Compass RoPE method is shown as Algorithm 1

## G. Transfer to a Hierarchical/Window-Based Backbone

We further evaluate whether the benefit of Compass-RoPE transfers beyond plain ViT to a hierarchical/window-based backbone. Specifically, we conduct multi-resolution evaluation on **Swin-Tiny** and compare the original **relative position bias (RPB)**, **Mix-RoPE**, and **Compass-RoPE**. Following the same protocol as the main text, the model is trained at $224^2$ and evaluated at multiple test resolutions without fine-tuning. For resolution changes, we simply rescale the RoPE magnitude by $s = \frac{r_{\text{test}}}{r_{\text{train}}}$. That is, the frequency magnitude is divided by $s$ when evaluating at resolution $r_{\text{test}}$. This keeps the frequency scale better aligned with the changed coordinate range under resolution shifts.

*Table 14.* Image generation results on DiT. Replacing the learnable absolute positional embedding with Compass-RoPE improves all reported metrics.

| Model | FID↓ | sFID↓ | IS↑ | Precision↑ | Recall↑ |
|---|---|---|---|---|---|
| DiT | 67.3 | 12.5 | 20.3 | 0.365 | 0.566 |
| DiT-CompassRoPE | **61.2** | **11.5** | **22.6** | **0.393** | **0.588** |

*Table 15.* Practical overhead comparison on ViT-Small. Compass-RoPE introduces negligible overhead compared with Mix-RoPE.

| Method | Throughput (img/s) ↑ | FLOPs (G) |
|---|---|---|
| Mix-RoPE | 3216.73 | 8.498 |
| Compass-RoPE | 3216.68 | 8.498 |

As shown in Table 13, Mix-RoPE degrades faster than the original RPB baseline at high resolutions, suggesting that RoPE frequencies in hierarchical/window-based architectures are still affected by axial bias. In contrast, Compass-RoPE shows better extrapolation behavior and remains stronger than the baselines in the high-resolution regime. This indicates that the advantage of improving directional isotropy is not restricted to plain ViT, but can also transfer to hierarchical/window-based transformers.

## H. Image Generation with DiT

We further evaluate whether the benefit of Compass-RoPE extends beyond discriminative vision tasks to image generation. Specifically, we replace the learnable absolute positional embedding in **DiT** with **Compass-RoPE**, while keeping the rest of the architecture and training setup unchanged. We report the standard generation metrics, including FID, sFID, IS, Precision, and Recall. As shown in Table 14, Compass-RoPE consistently improves DiT across all metrics. These results suggest that the advantage of improving directional isotropy is not limited to resolution extrapolation in discriminative settings, but can also transfer to generative transformers.

## I. Computational Overhead

We additionally report the practical overhead of Compass-RoPE on the **ViT-Small** setting used in the main paper. Although converting to polar coordinates and applying intra-head DFT technically change the computation compared with Mix-RoPE, the added cost is negligible in practice because both operations are lightweight and only applied inside the positional encoding module. Table 15 shows that the throughput difference is negligible and the FLOPs are identical to the baseline. These results indicate that Compass-RoPE introduces almost no practical computational overhead.

