# OpenReview forum: "Compass-RoPE: Isotropic Rotary Position Embeddings for Vision Transformers"
_ICML.cc/2026/Conference — ICML 2026 regular_

### Official Review · Reviewer_opCK · 2026-03-03

**Soundness:** 3
**Presentation:** 3
**Significance:** 3
**Originality:** 3
**Overall Recommendation:** 4
**Confidence:** 2

**Summary:**

This paper intends to discuss a central concept in improving position embeddings for vision transformers (ViTs) by addressing limitations in existing 2D Rotary Position Embeddings (RoPE). The authors explore the challenge of resolution extrapolation, where ViTs trained on fixed-resolution images often degrade in performance when tested on higher resolutions. They identify an axial spectral bias introduced by patch tokenization, which leads to anisotropic frequency orientations in learned 2D RoPE variants like Mix-RoPE, limiting their ability to capture diverse positional relations. To mitigate this, they propose Compass-RoPE, which parameterizes frequencies in polar coordinates with uniform angle initialization to promote isotropy and incorporates intra-head discrete Fourier transform (DFT) mixing to enhance angular expressiveness. Through experiments on classification, object detection and semantic segmentation tasks, they demonstrate that Compass-RoPE achieves superior and more stable performance under large resolution shifts compared to baselines.

**Compliance With Llm Reviewing Policy:**

Affirmed.

**Key Questions For Authors:**

1. The experiments mainly focus on the ViT-Small model. Have you evaluated Compass-RoPE on larger backbones such as ViT-Base or ViT-Large? Beyond ViT, have you tested it on other Transformer architectures (e.g., Swin Transformer)? If so, do the extrapolation gains exhibit a similar scaling trend, or do they diminish as model size increases?
2. Does converting Cartesian coordinates to polar coordinates, together with the intra-head DFT mixing, change the computational or time complexity? The paper does not analyze the additional overhead introduced by Compass-RoPE in terms of FLOPs, inference latency, or training time.
3. The performance of Compass-RoPE without DFT is worse than Mix-RoPE at smaller resolutions (Table 4). You hypothesize that the uniform distribution makes the "initial learning difficulty" harder. Could you elaborate on this mechanism, and did you experiment with less rigid angle initialization strategies to balance this trade-off without strictly requiring DFT mixing?

**Limitations:**

This work has no negative impact on society.

**Strengths And Weaknesses:**

**Paper Strengths:**

1. The analysis regarding why 2D RoPE frequencies become anisotropic during training is both novel and robust. Connecting the spectral bias of strided sampling in tokenization directly to anisotropic orientation learning in ROPE components  offers deep, fundamental insights.
1. The motivation is exceptionally clear, making an abstract frequency problem visual and easy to understand.
1. This paper explores the resolution robustness in ViTs issue that limits the scalability of modern architectures for reasoning. The experimental results show that the method has strong practical value and can be applied to practical applications in the future.

**Weaknesses:**

Please see **Key Questions For Authors**.

---

> ### Author Rebuttal · Authors · 2026-03-31
>
> Thank you for the positive feedback and for these important questions.
>
> **Q1. Larger backbones / other Transformer architectures.**
> We have additionally evaluated **ViT-Base** and **Swin-Tiny**. For **ViT-Base**, we train it at 224 resolution on ImageNet-100 for 200 epochs and test it at multiple resolutions. **Mix-RoPE also drops at high resolutions (worse than APE baseline at 720 resolution)**, which suggests that the anisotropy/extrapolation issue is **not limited to ViT-Small**. In contrast, **Compass-RoPE remains more stable in the high-resolution regime and achieves the best average performance**, indicating that our method continues to be effective on larger backbones.
>
> | Method |      224 |      512 |      672 |      720 |      AVG |
> |---|---------:|---------:|---------:|---------:|---------:|
> | ViT-Base (APE) |     73.7 |     74.9 |     69.8 |     68.3 |     71.7 |
> | ViT-Base (Mix-RoPE) |     76.4 |     77.9 |     70.3 |     67.0 |     72.9 |
> | ViT-Base (Compass-RoPE) | **77.7** | **78.2** | **74.7** | **73.7** | **76.1** |
>
> We train **Swin-Tiny** at 224 resolution on ImageNet-1K and test it on multiple resolutions. At high resolutions, **Mix-RoPE degrades even faster than the original RPB baseline**, which further indicates that RoPE frequencies in hierarchical/window-based architectures are still affected by axial bias. In contrast, **Compass-RoPE shows better extrapolation behavior and remains stronger than the baselines at high resolutions**, suggesting that the method also performs well on other Transformer architectures.
>
> | Method |      224 |      448 |       512 |      672 |      AVG |
> |---|---------:|---------:|----------:|---------:|---------:|
> | Swin-Tiny (RPB) |     81.2 |     77.3 |      75.1 |     69.3 |     75.7 |
> | Swin-Tiny (Mix-RoPE) |     **81.4** |     77.5 |      74.9 |     65.2 |     74.8 |
> | Swin-Tiny (Compass-RoPE) | **81.4** | **78.3** | **76.3** | **70.3** | **76.6** |
>
> **Q2. Computational overhead.**
> We agree that the original submission should have included an explicit overhead analysis. Converting to polar coordinates and applying intra-head DFT do change the computation compared with standard Mix-RoPE, but the added cost is negligible in practice because both operations are lightweight and only applied within the positional encoding module. To make this concrete, we add the following supplementary comparison on the **ViT-Small setting used in the main paper**, using **throughput** and **FLOPs** as metrics:
>
> | Method | Throughput (img/s) | FLOPs (G) |
> |---|---:|---:|
> | Mix-RoPE | 3216.73 | 8.498 |
> | Compass-RoPE | 3216.68 | 8.498 |
>
> As shown above, the throughput difference is negligible and the FLOPs are identical to the baseline, indicating that Compass-RoPE introduces almost no practical computational overhead.
>
> **Q3. Why is Compass-RoPE w/o DFT slightly worse than Mix-RoPE at smaller resolutions?**
>
> Our interpretation is a coverage–optimization trade-off. Because the patch-tokenized features already exhibit axial bias, their spectral energy is concentrated in a few dominant directions. Under such features, allowing RoPE frequencies to align toward these dominant directions—as Mix-RoPE does—makes attention learning easier near the training resolution. In contrast, Compass-RoPE w/o DFT initializes the frequency angles more uniformly, which prevents early concentration on those dominant directions and therefore makes optimization slightly harder. This explains why it is only 0.1 lower than Mix-RoPE near the training resolution, while still being substantially better under large resolution shifts (+4.2 at 720). In other words, Mix-RoPE benefits more from the biased training regime, but this comes at the cost of poorer extrapolation: because its frequencies are concentrated on only a few directions and have weak components elsewhere, it is less able to capture the changed coordinate distribution at OOD resolutions.
> Importantly, once DFT mixing is added, the full Compass-RoPE consistently outperforms the baselines across resolutions, showing that DFT effectively alleviates this trade-off by improving within-head angular expressiveness while preserving the extrapolation advantage. We have not yet systematically explored less rigid angle initialization strategies to balance this trade-off, but we agree that this is a promising direction for future work.

---

> > ### Author Rebuttal · Reviewer_opCK · 2026-04-04
> >
> > The results of ViT-Base greatly exceeded my expectations. Its performance shows a remarkably larger improvement over ViT-Tiny and ViT-Small than I had anticipated. I would appreciate it if the authors could further elaborate on the reasons behind this significant gain. At this stage, I am inclined to retain the current scores.

---

> > > ### Author Response · Authors · 2026-04-08
> > >
> > > Thank you for the follow-up. Our interpretation is that a better positional encoding essentially provides a stronger spatial prior / inductive bias, and such priors can be especially helpful in smaller-data regimes. This is also consistent with prior work: **ConViT** [1] reports that on **full ImageNet-1K**, ConViT-S improves over DeiT-S by only **1.5 points** (**81.4% vs. 79.9%**), whereas on only **10% of ImageNet-1K**, the gain becomes much larger—about **11.6 points** (**59.6% vs. 48.0%**). The paper explicitly concludes that the inductive bias is “most helpful on small datasets,” and further notes that larger models may benefit even more from a suitable prior.
> > >
> > > From this perspective, the larger gain we observe on **ViT-Base trained on ImageNet-100** is not necessarily anomalous; rather, it is consistent with the phenomenon that a stronger inductive bias can yield more visible improvements in smaller-data regimes. Since our current larger-backbone evidence is limited to **ImageNet-100**, we do not regard it as conclusive for full large-scale settings. If compute resources permit, we plan to further include **ViT-Base / ViT-Large results on full ImageNet-1K** in future work.
> > >
> > > **Reference**
> > >
> > > [1] *ConViT: Improving Vision Transformers with Soft Convolutional Inductive Biases*, ICML 2021.

---

### Official Review · Reviewer_xtvd · 2026-03-10

**Soundness:** 2
**Presentation:** 2
**Significance:** 2
**Originality:** 2
**Overall Recommendation:** 4
**Confidence:** 5

**Summary:**

This paper introduces Compass-RoPE, which improves the extrapolation ability of ViTs by improving on 2D-RoPE. The authors first observe that patchification (2D-strided convolution) distorts frequency content leading to an anisotropic optimization of RoPE frequency. In other words, the model will learn to (over)rely on certain frequencies, and if these frequencies change at extrapolation, performance will suffer.  The authors then introduce Compass-RoPE which fixes this issue by parameterizing frequency in polar form, spread angles uniformly, and applying an intra-head discrete fourier transform (DFT).

Through a controlled experiment, Compass-RoPE extrapolates to larger images better than baselines (by +8% over MixRoPE and +3.4% over absolute position encoding). The authors ablate to show roughly half of Compass-RoPE's gains are due to the intra-head DFT. And finally, the authors show that Compass-RoPE indeed learns a more uniform directional distribution.

**Compliance With Llm Reviewing Policy:**

Affirmed.

**Final Justification:**

Thanks to the authors for the new experiments. My concerns have been adequately addressed so I will increase my score to weak accept. For me, a stronger accept requires that these new experiments be run at the full ImageNet-1K scale, rather than ImageNet-100, which is smaller and more suited for analysis or ablations IMHO. I understand training baselines on ImageNet-1K may not be feasible during 1 week, yet I recommend running them before the camera-ready to fully convince the reader of the proposed method's superiority.

**Key Questions For Authors:**

Can the authors consider using ImageNet-HR [4], which is a high-resolution ImageNet test set that enables testing extrapolation without relying on upsampling?

**Limitations:**

I do not see limitations discussed, I suggest noting the model size (ViT-Small)

**Strengths And Weaknesses:**

Strengths:
- Image-size / resolution extrapolation is an important challenge that needs more research, current ViTs struggle
- Strong experimental setup, training all models from scratch using the DeiT III training recipe
- Nice analysis

Major weaknesses:
- Missing baselines: Spherical RoPE [1], LieRE [2], ComRoPE [3], LookHere [4]

In particular, Compass-RoPE sees a 10.8% accuracy drop when extrapolating from 224px to 720px on ImageNet-1K. LookHere sees a 3% accuracy drop when extrapolation from 224px to 768px on ImageNet-1K. The training recipes and model sizes differ between these results so they are not directly comparable, but IMHO these other methods must be considered in the manuscript before acceptance.

Minor weaknesses:
- some typos (including in the abstract)
- fig 1 could be improved for clarity

[1] A Circular Argument: Does RoPE need to be Equivariant for Vision?, NeurIPS 2025

[2] LieRE: Lie Rotational Positional Encodings, ICML 2025

[3] ComRoPE: Scalable and Robust Rotary Position Embedding Parameterized by Trainable Commuting Angle Matrices, CVPR 2025

[4] LookHere: Vision Transformers with Directed Attention Generalize and Extrapolate, NeurIPS 2024

---

> ### Author Rebuttal · Authors · 2026-03-31
>
> Thank you for pointing out these important baselines and for suggesting ImageNet-HR. We have now added supplementary comparisons to address both concerns.
>
> **(1) Missing recent baselines.**
>
> We additionally compare against the recent visual RoPE baselines Spherical RoPE\[1\], LieRE\[2\], and ComRoPE\[3\], as well as the non-RoPE method LookHere\[4\], under the same ImageNet-100 setting (train at 224 for 200 epochs, test at multiple resolutions):
>
> | Model | 224 | 320 | 448 |      512 | 672 | 720 |      AVG |
> |---|---:|---:|---:|---------:|--:|---:|---------:|
> | ViT-Spherical | 73.4 | 76.2 | 74.3 |     73.5  | 68.3 | 66.7 |     72.1 |
> | ViT-LieRE | 73.8 | 76.2 | 75.0 |     73.6 | 68.7 | 67.0 |     72.4 |
> | ViT-ComRoPE | 74.0 | 76.3 | 75.0 |     73.6 | 69.5 | 67.8 |     72.7 |
> | **ViT-Compass** | **74.4** | **77.2** | **76.3** | **74.3** | **69.6** | **67.8** | **73.3** |
> | ViT-LH | 73.7 | 76.6 | 76.2 |     75.5 | 72.0 | 70.3 |     74.1 |
> | **ViT-Compass-LH** | **74.6** | **76.7** | **76.9** | **76.0** | **72.4** | **70.4** | **74.5** |
>
> These results clarify two points. First, **among recent RoPE-family methods [1,2,3], Compass-RoPE is consistently the strongest across resolutions**, which directly supports our claim in the RoPE design space. Second, we agree that **LookHere [4] is stronger in absolute extrapolation** than Compass-RoPE alone. At the same time, the two methods are **complementary rather than redundant**: LookHere changes the attention pattern with a directed-attention design, whereas Compass-RoPE improves the directional allocation of learned RoPE frequencies. Because they act at different levels, they can be combined effectively; indeed, **Compass+LH outperforms either method alone**, showing that our method remains valuable even against a stronger non-RoPE baseline. This is also consistent with our paper’s core contribution: we target the anisotropic frequency failure mode of learned 2D RoPE caused by axial spectral bias, and improve directional coverage through polar parameterization and intra-head DFT mixing.
>
> **(2) ImageNet-HR evaluation.**
> Following your suggestion, we also evaluate the ImageNet-1K-trained models on **ImageNet-HR**, which avoids relying on upsampled validation images:
>
> | Model | 224 | 320 | 448 |        512 |  672 | 720 |      AVG |
> |---|---:|---:|---:|-----------:|--:|---:|---------:|
> | ViT-APE | 88.4 | 87.9 | 86.2 |       84.5 | 78.8 | 77.2 |     83.8 |
> | ViT-Mix | 88.4 | 89.7 | 88.5 |       86.6 | 77.4 | 72.0 |     83.8 |
> | ViT-Compass | 88.4 | **89.8** | **88.6** | **87.7**   | **80.4** | **77.6** | **85.4** |
>
> Compass-RoPE remains the best method on the extrapolation resolutions and achieves the best overall average. This suggests that the gains of Compass-RoPE are **not an artifact of evaluating on upsampled images**, but persist on a genuinely high-resolution test set as well.
>
> We have cited the above works, include direct comparisons with them and the ImageNet-HR experiments in our revised paper, and correct the typos you pointed out. We have also expanded the discussion of our method in relation to LookHere, focusing in particular on their complementarity and combined performance.
>
> **(3) Limitation of model size.**
> We agree that the current submission should discuss this limitation more explicitly, since the main controlled study is conducted on ViT-Small. We have added this point in the revised paper. To further address this concern, we now include supplementary results on **ViT-Base**, trained on **ImageNet-100 for 200 epochs at 224 resolution** and evaluated at multiple test resolutions:
>
> | Method | 224 | 512 | 672 | 720 | AVG |
> |---|---:|---:|---:|---:|---:|
> | ViT-Base (APE) | 73.7 | 74.9 | 69.8 | 68.3 | 71.7 |
> | ViT-Base (Mix-RoPE) | 76.4 | 77.9 | 70.3 | 67.0 | 72.9 |
> | ViT-Base (Compass-RoPE) | **77.7** | **78.2** | **74.7** | **73.7** | **76.1** |
>
> The same trend remains clear: **Mix-RoPE degrades at high resolutions and even underperforms APE at 720**, whereas **Compass-RoPE remains more stable in the high-resolution regime and achieves the best average performance**. These results suggest that our conclusion is **not limited** to model size (ViT-Small).
>
> **References**
>
> [1] *A Circular Argument: Does RoPE need to be Equivariant for Vision?, NeurIPS 2025*
>
> [2] *LieRE: Lie Rotational Positional Encodings, ICML 2025*
>
> [3] *ComRoPE: Scalable and Robust Rotary Position Embedding Parameterized by Trainable Commuting Angle Matrices, CVPR 2025*
>
> [4] *LookHere: Vision Transformers with Directed Attention Generalize and Extrapolate, NeurIPS 2024*

---

> > ### Author Rebuttal · Reviewer_xtvd · 2026-04-02
> >
> > Thanks to the authors for the new experiments. My concerns have been adequately addressed so I will increase my score to weak accept. For me, a stronger accept requires that these new experiments be run at the full ImageNet-1K scale, rather than ImageNet-100, which is smaller and more suited for analysis or ablations IMHO. I understand training baselines on ImageNet-1K may not be feasible during 1 week, yet I recommend running them before the camera-ready to fully convince the reader of the proposed method's superiority.

---

> > > ### Author Response · Authors · 2026-04-08
> > >
> > > Thank you very much for the positive feedback and for taking the time to carefully read our rebuttal. We are glad that the additional experiments helped address your concerns, and we sincerely appreciate your support and updated score.
> > >
> > > We agree that full ImageNet-1K validation would make the evidence stronger. In the camera-ready version, we will incorporate the new rebuttal experiments, including the comparisons with the listed recent baselines. We will also make our best effort to add ImageNet-1K experiments, if computation resources permit.

---

### Official Review · Reviewer_Ug6f · 2026-03-12

**Soundness:** 3
**Presentation:** 2
**Significance:** 2
**Originality:** 3
**Overall Recommendation:** 4
**Confidence:** 4

**Summary:**

This paper introduces Compass-RoPE, a novel positional embedding designed to improve resolution extrapolation in Vision Transformers (ViTs).
The authors identify that standard 2D RoPE methods suffer from an axial spectral bias, which leads to anisotropic learned frequencies that degrade generalization to unseen resolutions.
To address this, Compass-RoPE parameterizes frequencies in polar coordinates to explicitly decouple scale and angle, initializes angles uniformly, and applies intra-head discrete Fourier transform (DFT) mixing.
These modifications enforce isotropic directional coverage and enhance angular expressiveness, resulting in more stable and improved multi-resolution extrapolation performance.

**Compliance With Llm Reviewing Policy:**

Affirmed.

**Final Justification:**

I appreciate the authors' explanation regarding their compute constraints and the supplementary directional coverage analysis.
I acknowledge the hardware limitations, but I strongly urge the authors to explicitly state this validation scope limitation in the text and make every effort to include the full ImageNet-1K results for larger backbones in the camera-ready version to solidify the paper's claims.
Based on the clarifications provided during the rebuttal, I will maintain my current score.

**Key Questions For Authors:**

1. Does the axial spectral bias and the resulting extrapolation degradation hold true for larger capacity models like ViT-Base and ViT-Large?

2. What is the computational overhead (e.g., training throughput and inference latency) of the polar coordinate conversions and DFT mixing compared to the standard Mix-RoPE implementation?

3. Have you observed this frequency anisotropy in hierarchical/window-based Vision Transformers, and would Compass-RoPE be applicable and beneficial to them?

**Limitations:**

yes

**Strengths And Weaknesses:**

## Strengths

1. The paper offers a novel diagnosis of extrapolation failures in ViTs by tracing them back to an "axial spectral bias" caused by strided patch tokenization, which leads to anisotropic frequency distributions.

2. The proposed solution avoids heavy architectural modifications. By utilizing polar parameterization to decouple frequency scale and angle , combined with intra-head DFT mixing , the authors effectively force isotropic directional coverage.

## Weaknesses

1. The core multi-resolution extrapolation experiments rely exclusively on the ViT-Small model. It remains unverified whether higher-capacity models (ViT-Base or ViT-Large) suffer as severely from this bias or if they can naturally overcome it.

2. The method requires polar-to-Cartesian conversions and complex DFT matrix multiplications at every layer (Algorithm 1). The paper lacks profiling data regarding the actual impact on training throughput and inference latency.

---

> ### Author Rebuttal · Authors · 2026-03-31
>
> We thank the reviewer for these important questions. We add supplementary results on larger models, computational cost, and hierarchical/window-based backbones.
>
> **(1) Larger-capacity models.**
> We additionally evaluate **ViT-Base**. For **ViT-Base**, we train it at 224 resolution on ImageNet-100 for 200 epochs. We still observe axial spectral bias during training on the larger model, and **Mix-RoPE on ViT-Base also drops at high resolutions (worse than APE baseline at 720 resolution)**, showing that larger-capacity models are still affected by this issue. In contrast, **Compass-RoPE remains more stable in the high-resolution regime and achieves the best average performance**, indicating that our method also applies effectively to larger models.
>
> | Method                |      224 |      512 |      672 |      720 |      AVG |
> |-----------------------|---------:|---------:|---------:|---------:|---------:|
> | ViT-Base (APE)        |     73.7 |     74.9 |     69.8 |     68.3 |     71.7 |
> | ViT-Base (Mix-RoPE)   |     76.4 |     77.9 |     70.3 |     67.0 |     72.9 |
> | ViT-Base (Compass-RoPE) | **77.7** | **78.2** | **74.7** | **73.7** | **76.1** |
>
> **(2) Computational overhead.**
> We agree that the original submission should have included profiling. We therefore add the following comparison, measured on the ViT-Small setting used in the main paper. In practice, **Compass-RoPE introduces almost no extra parameter cost and only marginal throughput overhead** compared with Mix-RoPE.
>
> | Method |  Throughput (img/s)  | FLOPs (G) |
> |---|---------------------:|----------:|
> | Mix-RoPE |              3216.73 |     8.498 |
> | Compass-RoPE |              3216.68 |     8.498 |
>
>
> **(3) Hierarchical / window-based ViTs.**
> We additionally train **Swin-Tiny** at 224 resolution on ImageNet-1K and test it on higher resolutions. At high resolutions, **Mix-RoPE degrades even faster than the original RPB baseline**, which further suggests that RoPE frequencies in hierarchical/window-based architectures are still affected by axial bias. In contrast, **Compass-RoPE shows better extrapolation behavior and remains stronger than the baselines at high resolutions**, indicating that the method also transfers to this setting.
>
> | Method |      224 |      448 |       512 |      672 |      AVG |
> |---|---------:|---------:|----------:|---------:|---------:|
> | Swin-Tiny (RPB) |     81.2 |     77.3 |      75.1 |     69.3 |     75.7 |
> | Swin-Tiny (Mix-RoPE) |     **81.4** |     77.5 |      74.9 |     65.2 |     74.8 |
> | Swin-Tiny (Compass-RoPE) | **81.4** | **78.3** | **76.3** | **70.3** | **76.6** |
>
> Overall, these additional results show that (i) the anisotropy/extrapolation issue persists beyond ViT-Small, (ii) our method has negligible practical overhead, and (iii) Compass-RoPE is also effective on hierarchical/window-based transformers. We have included these additional experiments in the revised version.

---

> > ### Author Rebuttal · Reviewer_Ug6f · 2026-04-04
> >
> > Thank you for the rebuttal. While the Swin-Tiny and computational profiling adequately address my concerns on those fronts, the response regarding higher-capacity models remains fundamentally unresolved.
> >
> > Evaluating ViT-Base on ImageNet-100 is insufficient. The 73.7% baseline accuracy shows the model is underfitting. This fails to answer my core question: *Does a fully trained, data-rich large model naturally overcome the axial spectral bias?*
> >
> > To prove your claim, you must train ViT-Base (and ideally ViT-Large) on the **full ImageNet-1K dataset**. If the rebuttal window was too short for this compute, you must explicitly acknowledge this limitation in the paper and commit to adding the full IN-1K results in the camera-ready version.
> >
> > ***
> >
> > Without full-scale IN-1K validation for larger backbones, it is impossible to know if Compass-RoPE is actually necessary at scale, or if the bias naturally diminishes with capacity. This currently limits the paper's overall impact.

---

> > > ### Author Response · Authors · 2026-04-08
> > >
> > > Thank you for the clarification. We agree that our current **ViT-Base** result on **ImageNet-100** is not sufficient to fully determine whether axial spectral bias naturally diminishes for larger backbones trained at full **ImageNet-1K** scale.
> > >
> > > We would like to clarify the scope of our current evidence. In the main paper, all core experiments are conducted on **full ImageNet-1K**, and already establish the effectiveness of Compass-RoPE on **ViT-Small** under a strong **DeiT-III** [1] training setup. As also noted by reviewer **xtvd**, this is a strong baseline and an important strength of our paper. For larger backbones, we adopted ImageNet-100 as a supplementary setting because our current hardware cannot support stable full ImageNet-1K training of ViT-Base / ViT-Large under the DeiT-III recipe. Specifically, the DeiT-III configuration we follow relies on a global batch size of 2048 for stable training, while our lab currently has only 8×RTX 4090 GPUs, whose memory budget is insufficient to sustain this setup at the ViT-Base / ViT-Large scale on full ImageNet-1K.
> > >
> > > That said, we believe the additional evidence is still informative in two ways.
> > >
> > > First, we add a new analysis following the same protocol as **Table 5** in the main paper. Specifically, we measure the **effective directional coverage** of learned RoPE frequencies on **ViT-Base** by discretizing the orientation range $[0,\pi)$ into **36 bins**, so the **maximum possible value is 36**. A larger value indicates more uniform directional coverage.
> > >
> > > | Checkpoint | Training Data | Effective Direction Count ↑ |
> > > |---|---|---:|
> > > | Mix-RoPE | ImageNet-1K | 28.0 |
> > > | Compass-RoPE | ImageNet-100 | 35.9 |
> > >
> > > Although these two checkpoints are trained on different datasets, this result is still informative: the Mix-RoPE ViT-Base checkpoint trained on full ImageNet-1K **remains clearly anisotropic**, while Compass-RoPE reaches 35.9, which is very close to the theoretical maximum 36. This is consistent with our main finding that **Compass-RoPE learns a much more isotropic directional distribution**.
> > >
> > > Second, our ImageNet-100 ViT-Base experiment provides supporting evidence that Compass-RoPE can still mitigate this issue and improve extrapolation behavior on a larger backbone.
> > >
> > > We emphasize that the limitation is not about the method’s effectiveness on larger backbones, but about the current validation scope: our larger-model results are presently limited to ImageNet-100 training. We will clarify this more explicitly in the next version, and, if compute resources permit, further add full ImageNet-1K results for ViT-Base / ViT-Large in future work.
> > >
> > > **Reference**
> > >
> > > [1] DeiT III: Revenge of the ViT

---

### Official Review · Reviewer_bZAd · 2026-03-13

**Soundness:** 3
**Presentation:** 3
**Significance:** 2
**Originality:** 3
**Overall Recommendation:** 4
**Confidence:** 4

**Summary:**

This work found the limitation of existing Mix-RoPE: frequency components are anisotropic which makes it not robust to resolution change. In this work two technics are proposed to distribute frequency components more evenly which contributes to resolution robustness.

**Compliance With Llm Reviewing Policy:**

Affirmed.

**Key Questions For Authors:**

1. Patch tokenization can cause axial bias is a quite interesting observation. But it seems unclear to me why it is related with resolution change robustness. Different resolution should have the same axial bias, which shouldn't be a problem?
2. In the last paragraph of section 4. you mentioned the harm of anisotropic feature kernel theoretically. Do you have any exp results justifying this assumption?
3. In 5.2, why applying unitary DFT mixing could increase diversity of axis?
4. In table 1, Do you have any idea why the 320 resolution have hte highest metric? The model is trained with 224 then 320 would be slight OOD case?
5. In table 1, why does your method also drop with OOD inference input?
6. The biggest concern from my side: do you have applications other than resolution OOD? One can solve the resolution OOD with training with mixed resolution.

**Limitations:**

yes

**Strengths And Weaknesses:**

Strength: solid math, exps, nice derivation and figures, method proposed is effective both theoretically and empirically

Weakness:
1. Seems like resolution OOD is the main application case, which makes the contribution limited.
2. The method proposed doesn't solve the patchifying artifacts that tokenizer introduces, which can have broader impacts potentially.

---

> ### Author Rebuttal · Authors · 2026-03-31
>
> Thank you for the helpful questions. Our responses are as followed.
>
> **Q1.**
> As resolution changes, the dominant bias direction does not remain exactly identical; at higher resolutions it becomes more concentrated around 0 and $\pi/2$.
> More importantly, however, the key issue is not the bias direction itself, but that axial bias drives most learned RoPE frequencies toward a few dominant directions, leaving only small-magnitude components in the remaining directions. These weak components are effectively under-trained. For example, if a non-dominant directional frequency is only $\omega = 10^{-3}$, then on a $14\times14$ grid its maximum phase range is $10^{-3}\times14 = 0.014$, whereas on a $32\times32$ grid it becomes $10^{-3}\times32 = 0.032$. The newly exposed interval [0.014, 0.032] is therefore essentially unseen during training, creating an OOD phase regime.
> By contrast, along dominant directions, larger frequencies can already span a broader phase range within the training support, so extrapolation is relatively less harmful due to the periodicity of phase. This is consistent with our Sec. 4 analysis that resolution changes alter the relative-displacement distribution $\Delta$, making anisotropic frequencies less responsive along under-represented directions. A similar phenomenon has also been observed in NLP RoPE context extrapolation [1].
>
> **Q2.**
> Fig. 2 shows that axial bias indeed exists: patch tokenization makes the spectrum more axis-aligned. Fig. 3 and Table 5 then show that under such biased features, Mix-RoPE learns more anisotropic frequencies and covers a narrower set of directions than Compass-RoPE. Finally, Table 6 shows the consequence of this bias: anisotropic features distort the learned frequencies of Mix-RoPE, resulting in weaker modeling ability on under-represented directions—Mix-RoPE performs well on $0^\circ$ and $90^\circ$ shifts but degrades much more on $45^\circ$ and $135^\circ$. In contrast, Compass-RoPE remains more stable across all four directions.
>
> **Q3.**
> From the frequency perspective, DFT mixing linearly mixes the RoPE pairs within each head, so each transformed frequency component can contain multiple directional frequency patterns rather than only one original direction. This is exactly why Sec. 5.2 states that each mixed component can “nest multiple directional patterns.” As a result, each head has richer angular expressiveness and is less likely to collapse to a few dominant directions. Table 6 supports this interpretation: the full model consistently improves shift alignment over the w/o DFT variant across all four directions.
>
> **Q4.**
> This is a classic phenomenon in vision training and has been reported in prior work [2,3]. When going from 224 to 320, the model benefits from more visual details and a denser token grid, while the resolution OOD is still relatively small. At larger resolutions, the OOD begins to dominate and performance drops. This same pattern is also shown in Table 1 for multiple baseline methods, not only ours.
>
> **Q5. **
> As test resolution moves farther from 224, the token lattice, relative displacement distribution, and patch statistics all change, so some performance drop is expected. Our method mainly mitigates the OOD effect caused by changes in the **relative displacement distribution**, but it **cannot** remove **all** sources of OOD under large resolution shifts.
>
> **Q6.** Are there applications other than resolution OOD?
>
> Yes. Even if we do not consider OOD resolutions and look only at the **training resolution**, Table 1 already shows that Compass-RoPE outperforms the baseline methods at 224, so the benefit is not restricted to OOD inference only. Beyond the resolution-OOD setting, our paper also shows gains on **detection** and **segmentation**, and Table 7 further shows improved robustness under **combined rotation + resolution shifts**. In addition, we add an **image-generation** experiment by replacing the learnable APE in DiT with Compass-RoPE:
>
> | Model | FID↓ | sFID↓ | IS↑ | Precision↑ | Recall↑ |
> |---|---:|---:|---:|---:|---:|
> | DiT | 67.3 | 12.5 | 20.3 | 0.365 | 0.566 |
> | DiT-CompassRoPE | **61.2** | **11.5** | **22.6** | **0.393** | **0.588** |
>
> These results show that the utility of our method is not limited to resolution-OOD extrapolation.
> As for mixed-resolution training, it may be useful, but it brings more memory/compute cost (e.g., much larger GPU memory for high-resolution inputs such as 720×720), whereas our method can be trained once at low resolution and directly applied to higher resolutions at inference time.
>
> **References**
>
> [1] *Extending Context Window of Large Language Models via Positional Interpolation, NeurIPS 2024*
>
> [2] *Fixing the Train-Test Resolution Discrepancy, NeurIPS 2019*
>
> [3] *Training Data-Efficient Image Transformers & Distillation through Attention (DeiT), ICML 2021*

---

> > ### Author Rebuttal · Reviewer_bZAd · 2026-04-03
> >
> > I appreciate the additional experiment on DiT for image generation. The rebuttal has well addressed my questions and concerns. I'm happy to raise my score to accept.

---

> > > ### Author Response · Authors · 2026-04-08
> > >
> > > Thank you very much for the encouraging feedback and for taking the time to read our rebuttal carefully. We are glad that the additional DiT experiment helped address your concerns, and we sincerely appreciate your positive assessment. We are also grateful for your consideration of an updated score.

---

### Decision · Program_Chairs · 2026-04-30

**Decision:**

Accept (regular)

**Comment:**

Initially, reviewers appreciated the novel diagnosis of axial spectral bias in patch tokenization and the proposed solution. However, concerns remained regarding missing baselines, computational overhead, and generalization beyond the ViT-Small architecture. During the rebuttal, the authors provided extensive results, including successful applications to generative and hierarchical models, comparisons against newly requested baselines, and profiling demonstrating negligible overhead. While reviewers agree the method is technically sound, and the rebuttal was highly effective, the empirical validation is not bulletproof; specifically, it lacks a full ImageNet-1K training run on larger models (e.g., ViT-Base or ViT-Large). The AC agrees that the contributions are solid, but this limitation in the validation limits a recommendation of full acceptance. Therefore, the paper is recommended for weak acceptance.